# IMPROVING CONSISTENCY MODELS WITH GENERATOR-INDUCED FLOWS

## ABSTRACT

Consistency models imitate the multi-step sampling of score-based diffusion in a single forward pass of a neural network. They can be learned in two ways: consistency distillation and consistency training. The former relies on the true velocity field of the corresponding differential equation, approximated by a pre-trained neural network. In contrast, the latter uses a single-sample Monte Carlo estimate of this velocity field. The related estimation error induces a discrepancy between consistency distillation and training that, we show, still holds in the continuous-time limit. To alleviate this issue, we propose a novel flow that transports noisy data towards their corresponding outputs derived from the currently trained model – as a proxy of the true flow. Our empirical findings demonstrate that this approach mitigates the previously identified discrepancy. Furthermore, we present theoretical and empirical evidence indicating that our generator-induced flow surpasses dedicated optimal transport-based consistency models in effectively reducing the noise-data transport cost. Consequently, our method not only accelerates consistency training convergence but also enhances its overall performance.

## 1 INTRODUCTION

The large family of diffusion (Ho et al., 2020), score-based (Song et al., 2021; Karras et al., 2022), and flow models (Liu et al., 2023; Lipman et al., 2023) have emerged as state-of-the-art generative models for image generation. Since they are costly to use at inference time – requiring several neural function evaluations –, many distillation techniques have been explored (Salimans and Ho, 2022; Meng et al., 2023; Sauer et al., 2023). A most remarkable approach is *consistency models* (Song et al., 2023; Song and Dhariwal, 2024). Consistency models lead to high-quality one-step generators, that can be trained either by distillation of a pre-trained velocity field (*consistency distillation*), or as standalone generative models (*consistency training*) by approximating the velocity field through a one-sample Monte Carlo estimate.

The corresponding estimation error naturally induces a discrepancy between consistency distillation and training. While Song et al. (2023) hinted that it would resorb in the continuous-time limit, we show that this discrepancy remains both on loss functions' gradients and values. Interestingly, this discrepancy vanishes when the difference between the target velocity field and its Monte-Carlo approximation approaches zero. However, this is not the case with the independent coupling (IC) between data and noise used to construct the standard estimate. It is unclear how to improve this one-sample estimate without access to the true underlying diffusion model.

The approach we adopt in this paper to alleviate this issue consists in altering the velocity field – thereby changing the target flow – to reduce the variance of its one-sample estimator. One possible solution to this problem from the consistency and flow matching literatures (Pooladian et al., 2023; Dou et al., 2024) is to resort to optimal transport (OT) to learn on a deterministic coupling. However, due to the prohibitive cubic complexity of OT solvers (*e.g.* Hungarian matching algorithm), such methods need to be applied at the minibatch level. This incurs an OT approximation error (Fatras et al., 2021; Sommerfeld et al., 2019) and stochasticity of the data-noise coupling, thus not solving the consistency training issue.

In our approach, we propose to use the consistency model itself to construct trajectories. Indeed, the consistency model provides an approximation of the target flow via a single-step inference, and can thus be used as a proxy to reduce the expected deviation between the velocity field and its

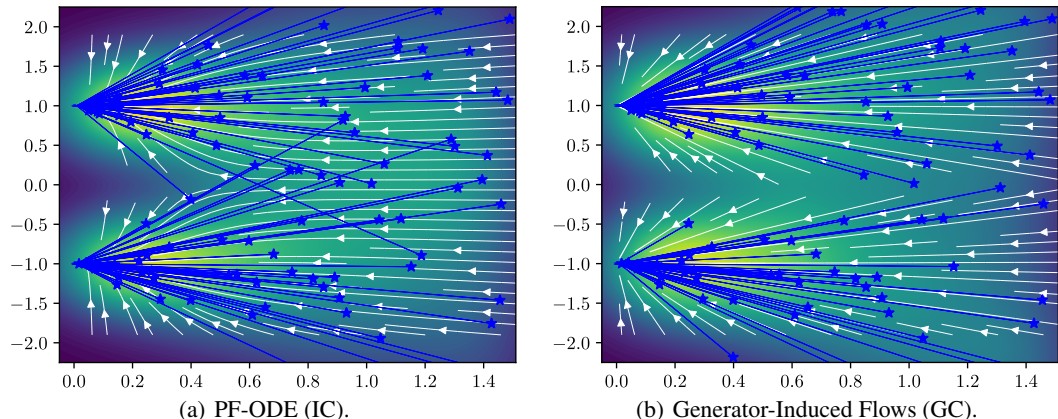

(a) PF-ODE (IC).  (b) Generator-Induced Flows (GC).

Figure 1: Comparison of the probability flow ODE (PF-ODE) and Generator-Induced Flows in a synthetic setting: target data is composed of two Diracs, and GC is computed with a closed-form generator. In the background, we observe the density of probability paths. White arrows are ODE trajectories associated to the velocity field. Blue lines are sample paths from IC in (a) (respectively GC in (b)). Trajectories start from random intermediate points ★. On this example, GC sample paths appear more aligned to the velocity field.

estimator. More precisely, from an intermediate point computed from an IC, we let the consistency model predict the corresponding endpoint, supposedly close to the data distribution. This predicted endpoint is coupled to the same original noise vector, defining a generator-induced coupling (GC). We show empirically that the resulting generator-induced flow presents favorable properties for training consistency models, in particular a reduced deviation between the velocity field and its estimator, besides reduced transport costs as supported by theoretical evidence. This can be observed in Figure 1. From this, we derive practical algorithms to train consistency models with generator-induced flows, leading to improved performance and faster convergence compared to standard and OT-based consistency models.

Let us summarize our contributions below.

- We prove that in the continuous time-limit consistency training and consistency distillation loss function converge to different pointwise values and we provide a closed-form expression of this discrepancy.

- We propose a novel type of flows that we denote *generator-induced flows*. It relies on generator-induced coupling (GC) that can be used to train a consistency model.

- We provide theoretical and empirical insights into the advantages of GC, notably on image datasets. We show that generator-induced flows have smaller discrepancy to consistency distillation than IC consistency training, and that they reduce data-noise transport costs.

- We derive practical ways to train consistency models with generator-induced flows in image generation benchmarks. Our approach based on a mixing strategy leads to faster convergence and improves the performance compared to the base model and OT-based approaches.

**Notation.** We consider an empirical data distribution $p_\star$ and a noise distribution $p_z$ (*e.g.* Gaussian), both defined on $\mathbb{R}^d$. We denote by $q$ a joint distribution of samples from $p_\star$ and $p_z$. We equip $\mathbb{R}^d$ with the dot product $\langle \mathbf{x}, \mathbf{y} \rangle = \mathbf{x}^\top \mathbf{y}$ and write $\|\mathbf{x}\| = \langle \mathbf{x}, \mathbf{x} \rangle^{1/2}$ for the Euclidean norm of $\mathbf{x}$. We use a distance function $\mathcal{D} \colon \mathbb{R}^d \times \mathbb{R}^d \to [0, \infty)$ to measure the distance between two points from $\mathbb{R}^d$. sg denotes the stop-gradient operator.

In consistency models, we consider diffusion processes of the form $\mathbf{x}_t = \mathbf{x}_\star + \sigma_t \mathbf{z}$, where $\mathbf{x}_\star \sim p_\star$, $\mathbf{z} \sim p_z$, and $\sigma_t$ is monotonically increasing for $t \in [0, T]$. We denote the distribution of $\mathbf{x}_t$ by $p(\mathbf{x}_t)$, or simply $p_t$. The conditional distributions or finite-dimensional joint distributions of $\mathbf{x}_t$'s are denoted similarly. When considering a discrete formulation with $N$ intermediate timesteps, we denote the intermediate points as $\mathbf{x}_{t_i} = \mathbf{x}_\star + \sigma_{t_i} \mathbf{z}$, where $t_i$ is strictly increasing for $i \in \{0, \ldots, N\}$,

with $t_0 = 0$ and $t_N = T$. The values of $\sigma_0$ and $\sigma_T$ are chosen to be sufficiently small and large, respectively, so that $p_0 \approx p_\star$ and $p_T \approx p(\sigma_T \mathbf{z})$.

## 2 CONSISTENCY MODELS: DISTILLATION VS TRAINING

In this section, we provide the required background on diffusion and consistency models (Sections 2.1 and 2.2), then discuss the discrepancy between consistency dillation and consistency training (Section 2.3) which we theoretically characterize in continous time.

### 2.1 FLOW AND SCORE-BASED DIFFUSION MODELS

Score-based diffusion models (Ho et al., 2020; Song et al., 2021) can generate data from noise via a multi-step process consisting in numerically solving either a stochastic differential equation (SDE), or equivalently an ordinary differential equation (ODE). Although SDE solvers generally exhibit superior sampling quality, ODEs have desirable properties. Most notably, they define a deterministic mapping from noise to data. Recently, Liu et al. (2023) and Lipman et al. (2023) generalize diffusion to flow models, which are defined by the following probability flow ODE (PF-ODE):

$$\mathrm{d}\mathbf{x} = \mathbf{v}_t(\mathbf{x})\,\mathrm{d}t, \tag{1}$$

where $\mathbf{v}_t(\mathbf{x}) = \mathbb{E}[\dot{\mathbf{x}}_t | \mathbf{x}_t = \mathbf{x}]$ is the velocity field. Note that $\dot{\mathbf{x}}_t$ is defined as the random variable $\dot{\mathbf{x}}_t = \frac{\mathrm{d}(\mathbf{x}_\star + \sigma_t \mathbf{z})}{\mathrm{d}t} = \dot{\sigma}_t z$, and is not to be confused with the time-derivative of the ODE, $\mathbf{v}_t$.

In the context of consistency models (Song et al., 2023; Song and Dhariwal, 2024), the most common choice is $\mathbf{v}_t(\mathbf{x}) = -\dot{\sigma}_t \sigma_t \nabla_\mathbf{x} \log p_t(\mathbf{x})\,\mathrm{d}t$, in particular the EDM formulation (Karras et al., 2022) where $\sigma_t = t$ and thus $\mathbf{v}_t^{\text{PF-ODE}}(\mathbf{x}) = -t\nabla_\mathbf{x} \log p_t(\mathbf{x})$. Here, $\nabla_\mathbf{x} \log p_t$, *a.k.a.* the score function, can be approximated with a neural network $\mathbf{s}_\phi(\mathbf{x}, t)$ (Vincent, 2011; Song and Ermon, 2019).

### 2.2 CONSISTENCY MODELS

Numerically solving an ODE is costly because it requires multiple expensive evaluations of the velocity function. To alleviate this issue, Song et al. (2023) propose training a *consistency model* $\boldsymbol{f}_\theta$, which learns the output map of the PF-ODE, *i.e.* its flow, such that:

$$\boldsymbol{f}_\theta(\mathbf{x}_t, \sigma_t) = \mathbf{x}_0, \tag{2}$$

for all $(\mathbf{x}_t, \sigma_t) \in \mathbb{R}^d \times [\sigma_0, \sigma_T]$ that belong to the trajectory of the PF-ODE ending at $(\mathbf{x}_0, \sigma_0)$.

Equation (2) is equivalent to *(i)* enforcing the boundary condition $\boldsymbol{f}_\theta(\mathbf{x}_0, \sigma_0) = \mathbf{x}_0$, and *(ii)* ensuring that $\boldsymbol{f}_\theta$ has the same output for any two samples of a single PF-ODE trajectory – the consistency property. *(i)* is naturally satisfied by the following model parametrization:

$$\boldsymbol{f}_\theta(\mathbf{x}_{t_i}, \sigma_{t_i}) = c_{\text{skip}}(t_i)\mathbf{x}_{t_i} + c_{\text{out}}(t_i)\boldsymbol{f}_\theta(\mathbf{x}_{t_i}, \sigma_{t_i}), \tag{3}$$

where $c_{\text{skip}}(0) = 1$, $c_{\text{out}}(0) = 0$, and $\boldsymbol{f}_\theta$ is a neural network. *(ii)* is achieved by minimizing the distance between the outputs of two same-trajectory consecutive samples using the consistency loss:

$$\mathcal{L}_{\text{CD}}(\theta) = \mathbb{E}_{q_{\text{I}}(\mathbf{x}_\star, \mathbf{z}), p(\mathbf{x}_{t_{i+1}} | \mathbf{x}_\star, \mathbf{z})} \Big[ \lambda(\sigma_{t_i})\mathcal{D}\Big( \text{sg}\big(\boldsymbol{f}_\theta(\mathbf{x}_{t_i}^\Phi, \sigma_{t_i})\big), \boldsymbol{f}_\theta(\mathbf{x}_{t_{i+1}}, \sigma_{t_{i+1}})\big) \Big], \tag{4}$$

where $(\mathbf{x}_\star, \mathbf{z})$ is sampled from the *independent* coupling $q_{\text{I}}(\mathbf{x}_\star, \mathbf{z}) = p_\star(\mathbf{x}_\star)p_z(\mathbf{z})$, $i$ is an index sampled uniformly at random from $\{0, 1, \ldots, N-1\}$, $\mathbf{x}_{t_{i+1}} = \mathbf{x}_\star + \sigma_{t_{i+1}}\mathbf{z}$, and $\mathbf{x}_{t_i}^\Phi$ is computed by discretizing the PF-ODE with the Euler scheme as follows:

$$\mathbf{x}_{t_i}^\Phi = \Phi(\mathbf{x}_{t_{i+1}}, t_{i+1}) = \mathbf{x}_{t_{i+1}} + (t_i - t_{i+1})\mathbf{v}_{t_{i+1}}^{\text{PF-ODE}}(\mathbf{x}_{t_{i+1}}). \tag{5}$$

This loss can be directly used to distill a score model into $\boldsymbol{f}_\theta$.

In the case of consistency training, Song et al. (2023) circumvent the lack of a score function by noting that $\mathbf{v}_{t_{i+1}}^{\text{PF-ODE}}(\mathbf{x}) = \mathbb{E}[\dot{\mathbf{x}}_{t_{i+1}} | \mathbf{x}_{t_{i+1}} = \mathbf{x}]$. In light of this, the intractable $\mathbf{x}_{t_i}^\Phi$ is replaced by its single-sample Monte Carlo estimate $\mathbf{x}_{t_i}$, resulting in:

$$\mathcal{L}_{\text{CT}}(\theta) = \mathbb{E}_{q_{\text{I}}(\mathbf{x}_\star, \mathbf{z}), p(\mathbf{x}_{t_i}, \mathbf{x}_{t_{i+1}} | \mathbf{x}_\star, \mathbf{z})} \Big[ \lambda(\sigma_{t_i})\mathcal{D}\Big( \text{sg}\big(\boldsymbol{f}_\theta(\mathbf{x}_{t_i}, \sigma_{t_i})\big), \boldsymbol{f}_\theta(\mathbf{x}_{t_{i+1}}, \sigma_{t_{i+1}})\big) \Big]. \tag{6}$$

### 2.3 DISCREPANCY BETWEEN CONSISTENCY TRAINING AND DISTILLATION AND VELOCITY FIELD ESTIMATION

Naturally, replacing $\mathbf{v}_t^{\text{PF-ODE}}$ by its single-sample estimate $\dot{\mathbf{x}}_t$ makes consistency training deviate from consistency distillation in discrete time. Still, Song et al. (2023, Theorems 2 and 6) suggest that this discrepancy disappears in continuous time since $\mathcal{L}_{\text{CT}}(\theta) = \mathcal{L}_{\text{CD}}(\theta) + o(1/N)$ and the corresponding gradients are equal in some cases. Without disproving these results, we find that scaling issues and lack of generality soften their claim of a closed gap between consistency training and distillation.

Indeed, we provide in the following theorem a thorough theoretical comparison of $\mathcal{L}_{\text{CT}}$ and $\mathcal{L}_{\text{CD}}$. We first prove that they converge to different values in the continuous-time limit. The difference is captured by a regularization term that depends on the discrepancy between the velocity field and its estimate. Moreover, we show that the limits of the scaled gradients do not coincide in the general case, leaving the (asymptotic) quadratic loss as essentially the only case where the two limits happen to coincide. The proof can be found in Appendix A.1.

**Theorem 1** (**Discrepancy between consistency distillation and consistency training objectives**). *Assume that the distance function is given by $\mathcal{D}(\mathbf{x}, \mathbf{y}) = \varphi(\|\mathbf{x} - \mathbf{y}\|)$ for a continuous convex function $\varphi : [0, \infty) \to [0, \infty)$ with $\varphi(x) \sim Cx^\alpha$ as $x \to 0^+$ for some $C > 0$ and $\alpha \geq 1$, and that the timesteps are equally spaced, i.e., $t_i = \frac{iT}{N}$. Furthermore, assume that the Jacobian $\frac{\partial \mathbf{f}_\theta}{\partial \mathbf{x}}$ does not vanish identically. Then the following assertions hold:*

  *(i) The scaled consistency losses $N^\alpha \mathcal{L}_{\text{CD}}(\theta)$ and $N^\alpha \mathcal{L}_{\text{CT}}(\theta)$ converge as $N \to \infty$. Moreover, the minimization objectives corresponding to these limiting scaled consistency losses are not equivalent, and their difference is given by:*
$$\lim_{N \to \infty} N^\alpha \left[ \mathcal{L}_{\text{CT}}(\theta) - \mathcal{L}_{\text{CD}}(\theta) \right] = CT^{\alpha-1} \mathcal{R}(\theta), \tag{7}$$

  *where $\mathcal{R}(\theta)$ is defined by*
$$\mathcal{R}(\theta) = \int_0^T \lambda(\sigma_t) \mathbb{E} \left[ \|\partial_{\text{CT}} \mathbf{f}_\theta\|^\alpha - \|\partial_{\text{CD}} \mathbf{f}_\theta\|^\alpha \right] \, \mathrm{d}t \tag{8}$$

  *and satisfies $\mathcal{R}(\theta) > 0$, with*
$$\partial_{\text{CT}} \mathbf{f}_\theta = \frac{\partial \mathbf{f}_\theta}{\partial \sigma}(\mathbf{x}_t, \sigma_t) \dot{\sigma}_t + \frac{\partial \mathbf{f}_\theta}{\partial \mathbf{x}}(\mathbf{x}_t, \sigma_t) \cdot \dot{\mathbf{x}}_t, \tag{9}$$
$$\partial_{\text{CD}} \mathbf{f}_\theta = \frac{\partial \mathbf{f}_\theta}{\partial \sigma}(\mathbf{x}_t, \sigma_t) \dot{\sigma}_t + \frac{\partial \mathbf{f}_\theta}{\partial \mathbf{x}}(\mathbf{x}_t, \sigma_t) \cdot \mathbf{v}_t(\mathbf{x}_t). \tag{10}$$

  *In particular, if $\alpha = 2$,*
$$\mathcal{R}(\theta) = \int_0^T \lambda(\sigma_t) \mathbb{E} \left[ \left\| \frac{\partial \mathbf{f}_\theta}{\partial \mathbf{x}}(\mathbf{x}_t, \sigma_t) \cdot (\dot{\mathbf{x}}_t - \mathbf{v}_t(\mathbf{x}_t)) \right\|^2 \right] \, \mathrm{d}t. \tag{11}$$

  *(ii) The scaled gradient $N^{\alpha-1} \nabla_\theta \mathcal{L}_{\text{CD}}(\theta)$ and $N^{\alpha-1} \nabla_\theta \mathcal{L}_{\text{CT}}(\theta)$ converge as $N \to \infty$. Moreover, if $\alpha \neq 2$, then their respective limits are not identical as functions of $\theta$:*
$$\lim_{N \to \infty} N^{\alpha-1} \nabla_\theta \mathcal{L}_{\text{CT}}(\theta) \neq \lim_{N \to \infty} N^{\alpha-1} \nabla_\theta \mathcal{L}_{\text{CD}}(\theta). \tag{12}$$

This theorem reveals that the optimization problems of consistency training and distillation differ not only in discrete time but also in continuous time. It even highlights a discrepancy between, firstly, the limiting gradients in continuous time—although they are equal for $\alpha = 2$—and, secondly, the gradients of the limiting losses, which differ because of $\mathcal{R}(\theta)$, even when $\alpha = 2$.

This analysis also shows the importance of employing probability paths whose sample path derivatives $\dot{\mathbf{x}}_t$ are aligned with the velocity field $\mathbf{v}_t(\mathbf{x}_t)$. Notably, if a diffusion process $\mathbf{x}_t$ satisfies $\dot{\mathbf{x}}_t = \mathbf{v}_t(\mathbf{x}_t)$, we have $\mathcal{R}(\theta) = 0$ and equal gradients for all $\alpha \geq 1$. Hence, for such $\mathbf{x}_t$, consistency training and consistency distillation would be reconciled both in discrete time and in the continuous-time limit.

However, it is unclear how to directly improve the single-sample estimation $\dot{\mathbf{x}}_t$ of $\mathbf{v}_t(\mathbf{x}_t)$. In particular, increasing the number of samples per point $\mathbf{x}_t$ to reduce its variance is not tractable, as it requires sampling from the inverse diffusion process $p(\mathbf{x}_\star | \mathbf{x}_t)$. Therefore, we adopt an alternative approach to alleviate the discrepancy identified in this section, which consists of altering the velocity field – thereby changing the target flow – to reduce the variance of its one-sample estimator. This approach is reminiscent of recent work tackling the data-noise coupling that we discuss in the following section.

## 3 REDUCING THE DISCREPANCY WITH DATA-NOISE COUPLING

**Beyond independent coupling (IC).** From Section 2.2, it appears that $\dot{\mathbf{x}}_t$ is computed through an IC $q_{\mathrm{I}} = p_\star(\mathbf{x}_\star)p_z(\mathbf{z})$ of data and noise, in a similar fashion to flow matching (Lipman et al., 2023; Kingma and Gao, 2024). Making correlated choices of data and noise beyond IC could then help align $\dot{\mathbf{x}}_t$ and $\mathbf{v}_t(\mathbf{x}_t)$, thereby resolving the discrepancy from the previous section. This question has been tackled, although with different motivation, by recent work.

The reliance on IC in consistency and flow models is increasingly recognized as a limiting factor. Recent advancements suggest that improved coupling mechanisms could enhance both training efficiency and the quality of generated samples in flow matching (Liu et al., 2023; Pooladian et al., 2023) and diffusion models (Li et al., 2024). By reducing the variance in gradient estimation, enhanced coupling can accelerate training. Additionally, improved coupling could decrease transport costs and straighten trajectories, yielding better-quality samples. In a different context, ReFlow (Liu et al., 2023) leverages couplings provided by the ODE solver in a flow framework, and demonstrate that it reduced transport costs.

**Couplings based on optimal transport (OT) solvers.** OT is a particularly appealing solution for our alignment problem. Indeed, if we consider a quadratic cost and distributions with bounded supports, OT is a no-collision transport map (Nurbekyan et al., 2020), *i.e.* $\mathbf{x}_t$ can be sampled by a unique pair of points $(\mathbf{x}_\star, \mathbf{z})$. Thus $\dot{\mathbf{x}}_t = \mathbf{v}_t(\mathbf{x}_t)$, implying $\mathcal{R}(\theta) = 0$ in Theorem 1. Several approaches have precisely targeted the reduction of transport cost in flow and consistency models.

Pooladian et al. (2023) have more directly explored OT coupling within the framework of flow matching models. They show that deterministic and non-crossing paths enabled by OT with infinite batch size lowers the variance of gradient estimators. Experimentally, they assess the efficacy of OT solvers, such as Hungarian matching and Sinkhorn algorithms, in coupling batches of noise and data points. Dou et al. (2024) have successfully adopted this approach in consistency models. However, due to the prohibitive cubic complexity of OT solvers, OT has to be applied by minibatch for matching samples $(\mathbf{x}_\star, \mathbf{z})$. Besides an OT approximation error, this incurs the loss of the no-collision property, making $\mathcal{R}(\theta)$ non-zero in real use-cases. Another line of works using OT tools with score-based models relies on the Schrödinger Bridge formulation (De Bortoli et al., 2021; Shi et al., 2023; Korotin et al., 2024; Tong et al., 2024), which has mostly proven benefits on transfer tasks.

**Our approach.** In this paper, we use a consistency model as a proxy of the flow of a diffusion process to reduce transport costs. Our method does not rely on an iterative procedure but rather on a mixing procedure during a single training, alleviating error accumulation. While not fully solving the alignment issue either, we will show that our method present reduced transport costs and better alignment than dedicated OT-based methods.

## 4 CONSISTENCY MODELS WITH GENERATOR-INDUCED FLOWS

Here, we introduce our method, denoted as Generator-Induced Flows, which relies on a Generator-Induced Coupling (GC). It consists of capitalizing on the consistency model's ability to approximate the diffusion flow in an affordable manner – during the training of the consistency model itself. This approach not only allows reducing the data-noise transport cost but also narrows the gap between consistency distillation and consistency training while maintaining a low computational overhead.

### 4.1 GENERATOR-INDUCED COUPLING (GC): DEFINITION AND TRAINING LOSS

The solution proposed in this work involves harnessing the consistency model to create a novel form of coupling. The idea is to leverage the properties and accumulated knowledge within the consistency model itself, $\boldsymbol{f}_\theta$, to construct pairs of points. To achieve this, we first sample an intermediate point, which is done as usual by sampling $\mathbf{x}_\star \sim p_\star$ and $\mathbf{z} \sim p_z$ using the IC between the two distributions, and then predict the initial data point $\hat{\mathbf{x}}_{t_i}$ via the consistency model:

$$(\mathbf{x}_\star, \mathbf{z}) \sim q_{\mathrm{I}}, \qquad \mathbf{x}_{t_i} = \mathbf{x}_\star + \sigma_{t_i}\mathbf{z}, \qquad \hat{\mathbf{x}}_{t_i} = \mathrm{sg}(\boldsymbol{f}_\theta(\mathbf{x}_{t_i}, \sigma_{t_i})). \qquad (13)$$

Although $\hat{\mathbf{x}}_{t_i}$ depends on the timestep $t_i$, it is important to note that it (supposedly) follows the distribution $p_0$. This $\hat{\mathbf{x}}_{t_i}$ is coupled with $\mathbf{z}$, thereby defining our *generator-induced coupling* (GC) $q$,

---

**Algorithm 1:** Training of consistency models with generator-induced trajectories.

**Input:** Randomly initialized consistency model $\boldsymbol{f}_\theta$, number of timesteps $N$, noise schedule $\sigma_{t_i}$, loss weighting $\lambda(\cdot)$, learning rate $\eta$, distance function $\mathcal{D}$, noise distribution $p_z$.

**Output:** Trained consistency model $\boldsymbol{f}_\theta$.

```
1 while not converged do
2     x⋆ ∼ p⋆,  z ∼ pz ;           // batch of real data and noise vectors
3     i ∼ multinomial(p(σt₀), ..., p(σtₙ)) ;        // sampling timesteps
4     xtᵢ ← x⋆ + σtᵢ z ;                      // IC intermediate points
5     x̂tᵢ ← sg(fθ(xtᵢ, σtᵢ)) ;       // endpoint prediction from the model
6     x̃tᵢ ← x̂tᵢ + σtᵢ z,  x̃tᵢ₊₁ ← x̂tᵢ + σtᵢ₊₁ z ;     // GC intermediate points
7     L(θ) = λ(σtᵢ)D(sg(fθ(x̃tᵢ, σtᵢ)), fθ(x̃tᵢ₊₁, σtᵢ₊₁)) ;    // consistency loss
8     θ ← θ − η∇θL(θ) ;              // back-propagate consistency loss
```

---

which we use to construct the pair of points $(\tilde{\mathbf{x}}_{t_i}, \tilde{\mathbf{x}}_{t_{i+1}})$:

$$(\hat{\mathbf{x}}_{t_i}, \mathbf{z}) \sim q, \qquad \tilde{\mathbf{x}}_{t_i} = \hat{\mathbf{x}}_{t_i} + \sigma_{t_i}\mathbf{z}, \qquad \tilde{\mathbf{x}}_{t_{i+1}} = \hat{\mathbf{x}}_{t_i} + \sigma_{t_{i+1}}\mathbf{z}. \qquad (14)$$

Then we define the consistency training loss for GC, with the overall training procedure outlined in Algorithm 1:

$$\mathcal{L}_{\mathrm{GC}}(\theta) = \mathbb{E}_{q(\hat{\mathbf{x}}_{t_i}, \mathbf{z}|\theta), p(\tilde{\mathbf{x}}_{t_i}, \tilde{\mathbf{x}}_{t_{i+1}}|\hat{\mathbf{x}}_{t_i}, \mathbf{z})}\Big[\lambda(\sigma_{t_i})\mathcal{D}\Big(\mathrm{sg}(\boldsymbol{f}_\theta(\tilde{\mathbf{x}}_{t_i}, \sigma_{t_i})), \boldsymbol{f}_\theta(\tilde{\mathbf{x}}_{t_{i+1}}, \sigma_{t_{i+1}})\Big)\Big]. \qquad (15)$$

**Generator-induced trajectories satisfy the boundary conditions of diffusion processes.** We note the two following important properties of the distribution of $\tilde{\mathbf{x}}_t$:

$$p(\tilde{\mathbf{x}}_0) = p(\mathbf{x}_0) \approx p_\star \qquad \text{and} \qquad p(\tilde{\mathbf{x}}_T) \approx p(\mathbf{x}_T) \approx p(\sigma_T\mathbf{z}) \qquad (16)$$

The first property is achieved thanks to the boundary condition of the consistency model (*c.f.* Section 2.1), and the second property by construction of the diffusion process which ensures that the noise magnitude is significantly larger than $\hat{\mathbf{x}}_{t_i}$ for large $t$. However, for the timesteps $t \in (0, T)$ the marginal distributions $p(\mathbf{x}_t)$ and $p(\tilde{\mathbf{x}}_t)$ do not necessarily coincide. An example of the impact of this distribution shift can be observed in Figure 4(b).

## 4.2 PROPERTIES OF GENERATOR-INDUCED FLOWS

Here, we present some properties of Generator-Induced Flows that motivate them for training consistency models.

### 4.2.1 REDUCING $\mathcal{R}(\theta)$ WITH GC

In Theorem 1, we proved that the continuous-time consistency training objective decomposes into the sum of the consistency distillation objective and a regularizer term: $\mathcal{L}_{\mathrm{CT}}(\theta) = \mathcal{L}_{\mathrm{CD}}(\theta) + \mathcal{R}(\theta)$. Here, we study a proxy term for $\mathcal{R}(\theta)$ that is easier to calculate:

$$\tilde{\mathcal{R}}_t = \mathbb{E}\left[\left\|\dot{\mathbf{x}}_t - \mathbf{v}_t(\mathbf{x}_t)\right\|^2\right]. \qquad (17)$$

This quantity measures the expected distance between the true velocity field and its one-sample Monte Carlo estimate. Note that $\tilde{\mathcal{R}}_{t,\mathrm{GC}}$ depends on the endpoint predictor,

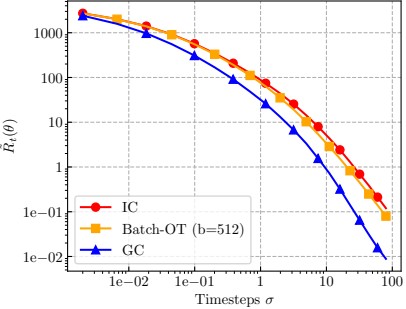

Figure 2: Comparison of $\tilde{\mathcal{R}}_{\mathrm{IC}}$, $\tilde{\mathcal{R}}_{\text{batch-OT}}$, and $\tilde{\mathcal{R}}_{\mathrm{GC}}$ on CIFAR-10. GC exhibits lower values of this quantity for all $\sigma_t$.

a consistency model, which impacts both probability paths and velocity fields. Our goal is to theoretically and empirically compare $\tilde{\mathcal{R}}_{t,\mathrm{IC}}$, $\tilde{\mathcal{R}}_{\text{batch-OT}}$, and $\tilde{\mathcal{R}}_{t,\mathrm{GC}}$ in order to demonstrate that GC does lead to a smaller discrepancy term than IC.

In the following theorem, proved in Appendix A.2, we show that $\tilde{\mathcal{R}}_t$ decays faster for GC than for IC.

**Theorem 2.** *Assume that the data distribution contains more than a single point. Also, assume that the generator-induced coupling between the predicted data point $\hat{\mathbf{x}}_t$ and noise $\mathbf{z}$ is computed via an ideal consistency model $\mathring{\boldsymbol{f}}$, i.e., the flow of the PF-ODE. Then, as $t \to \infty$,*

$$\mathring{\mathcal{R}}_{t,\mathrm{GC}} \ll \tilde{\mathcal{R}}_{t,\mathrm{IC}}. \qquad (18)$$

**Empirical validation.** Evaluating $\tilde{\mathcal{R}}_t$ requires computing the difference between the sample path derivative and the velocity field. In the EDM setting, this difference can be approximated using a denoiser. Indeed, $\dot{\mathbf{x}}_t = \mathbf{z}$ and $\mathbf{v}_t(\mathbf{x}_t) = \mathbb{E}[\dot{\mathbf{x}}_t|\mathbf{x}_t] = \mathbb{E}[\mathbf{z}|\mathbf{x}_t] = \mathbb{E}[\frac{\mathbf{x}_t-\mathbf{x}_\star}{t}|\mathbf{x}_t] = \frac{1}{t}(\mathbf{x}_t - \boldsymbol{D}_\star(\mathbf{x}_t, t))$ with an optimal denoiser $\boldsymbol{D}_\star$, which then can be replaced by a denoiser network $\boldsymbol{D}_\phi$ to yield an approximation for the difference: $\dot{\mathbf{x}}_t - \mathbf{v}_t(\mathbf{x}_t) \approx \mathbf{z} - \frac{1}{t}(\mathbf{x}_t - \boldsymbol{D}_\phi(\mathbf{x}_t, t))$. Since IC, batch-OT, and GC can define different $p_t$'s and $\mathbf{v}_t$'s, we train a different denoiser $\boldsymbol{D}_\phi$ for each coupling. In Figure 2, we report the results from the comparison of the two proxy terms on CIFAR-10. We observe that $\tilde{\mathcal{R}}_{t,\text{GC}} < \tilde{\mathcal{R}}_{t,\text{batch-OT}} < \tilde{\mathcal{R}}_{t,\text{IC}}$ and that the gap increases with $t$, corroborating our theoretical findings.

### 4.2.2 REDUCING TRANSPORT COST WITH GC

Here, we investigate the average transport cost between the noise $\mathbf{z} \sim p_z$ and the predicted data point $\hat{\mathbf{x}} \sim p_\star$ as a measure of the efficiency of the data-noise coupling. Recall that the diffusion process is given by $\mathbf{x}_t = \mathbf{x}_\star + \sigma_t \mathbf{z}$. Then, for a consistency model $\boldsymbol{f}$ satisfying the boundary condition $\boldsymbol{f}(\mathbf{x}_0, \sigma_0) = \mathbf{x}_0$, we define the function $c(t)$ as:

$$c(t) = \mathbb{E}_{q_\mathrm{I}(\mathbf{x}_\star, \mathbf{z})}\left[\left\|\boldsymbol{f}(\mathbf{x}_t, \sigma_t) - \mathbf{z}\right\|^2\right]. \tag{19}$$

$c(0) = \mathbb{E}_{q_\mathrm{I}(\mathbf{x}_\star, \mathbf{z})}[\|\mathbf{x}_\star - \mathbf{z}\|^2]$ and $c(t)$ represent the respective transport cost of IC and GC. We show below, with proofs in Appendix A.3, that $c(t)$ is decreasing for $\sigma_t$ close to zero and for $\sigma_t$ large.

**Lemma 1** (**Transport cost of GC coupling**). *Assume that $\boldsymbol{f}$ is a continuously differentiable function representing the ground-truth consistency model, i.e. the flow of the PF-ODE induced by the diffusion process $\mathbf{x}_t$. Define $\mathbf{w}_t = \mathbf{z} - \mathbb{E}[\mathbf{z}|\mathbf{x}_t] = \frac{1}{\sigma_t}(\dot{\mathbf{x}}_t - \mathbb{E}[\dot{\mathbf{x}}_t \mid \mathbf{x}_t])$. Then:*

$$c'(t) = -2\dot{\sigma}_t \mathbb{E}\left[\left\langle \frac{\partial \boldsymbol{f}}{\partial \mathbf{x}}(\mathbf{x}_t, \sigma_t) \cdot \mathbf{w}_t, \mathbf{w}_t \right\rangle\right]. \tag{20}$$

**Corollary 1** (**Decreasing transport cost of GC coupling in $t \to 0^+$**). *There exists a $t_* > 0$ such that for all $t \in [0, t_*]$, the derivative of $c(t)$ takes the form $c'(t) = -2\dot{\sigma}_t a_t$ with $a_t > 0$. Hence for $\dot{\sigma}_t$ positive, the cost is decreasing. In particular, in the EDM setting where $\sigma_t = t$, $c(t)$ is decreasing for small $t$.*

The proof of this corollary proceeds by noting that for $t = 0$, the consistency model $\boldsymbol{f}_\theta(\mathbf{x}, t)$ is an identity function, its Jacobian is an identity matrix, and thus $a_t = \mathbb{E}[\|\mathbf{w}_t\|^2]$. Using the continuity of Jacobian elements and invoking intermediate value theorem on $a_t$ concludes the proof.

**Corollary 2** (**Decreasing transport cost of GC coupling in $t \approx t_{\max}$**). *Assume that the consistency model $\boldsymbol{f}_\theta(x, \sigma)$ is a scaling function $\boldsymbol{f}_\theta(\mathbf{x}, \sigma_t) = \frac{\sigma_0}{\sigma_t}\mathbf{x}$. Then, we have $c'(t) = -\frac{2\dot{\sigma}_t \sigma_0}{\sigma_t}\mathbb{E}[\|\mathbf{w}_t\|^2]$. In particular, $c(t)$ is decreasing whenever $\sigma_t$ is increasing.*

We note that, while the assumption of the consistency model being a scaling function is strong, it nonetheless bears some degree of truth for $t \approx t_{\max}$, see Lemma 3 of Appendix A.

**Toy example.** Let us consider a one-dimensional toy example where $\mathbf{x}_\star \sim \mathcal{N}(0, \sigma_\star^2)$ with $\sigma_\star \geq 0$ and $\mathbf{z} \sim \mathcal{N}(0, 1)$ are independent. Also, we assume $\sigma_0 = 0$ for the sake of simplicity. In this case, the marginal law of $\mathbf{x}_t$ is also Gaussian with $p_t = \mathcal{N}(0, \sigma_\star^2 + \sigma_t^2)$, so the vector field for the diffusion process $\mathbf{x}_t$ is calculated as $\mathbf{v}_t(\mathbf{x}) = -\dot{\sigma}_t \sigma_t \nabla_\mathbf{x} \log p_t(\mathbf{x}) = \frac{\dot{\sigma}_t \sigma_t}{\sigma_\star^2 + \sigma_t^2}\mathbf{x}$. Then, the corresponding target diffusion flow and the transport cost function are given by:

$$\boldsymbol{f}(\mathbf{x}, \sigma_t) = \frac{\sigma_\star}{\sqrt{\sigma_\star^2 + \sigma_t^2}}\mathbf{x} \qquad \text{and} \qquad c(t) = \sigma_\star^2 + 1 - \frac{2\sigma_\star \sigma_t}{\sqrt{\sigma_\star^2 + \sigma_t^2}}. \tag{21}$$

We note that $\boldsymbol{f}(\mathbf{x}, \sigma_t)$ is indeed a scaling function which is asymptotically proportional to $\frac{\mathbf{x}}{\sigma_t}$ for large $t$, and $c(t)$ is decreasing in $t$ for $t > 0$.

**Experimental validation.** As stressed in Section 3, a line of works has brought evidence that reducing the transport cost between noise and data distributions could fasten the training and help produce better samples. We compare the quadratic transport costs involved in IC, batch-OT (Pooladian et al., 2023; Dou et al., 2024), and GC (resp. $c(0)$, $c_{\text{OT}}(0)$, and $c(t)$). Results are presented in Figure 3. Interestingly, GC reduces transport cost more than batch-OT on CIFAR-10 because batch OT is tied to the batch data points $\mathbf{x}_t$ whereas our computed $\hat{\mathbf{x}}_t$ are not.

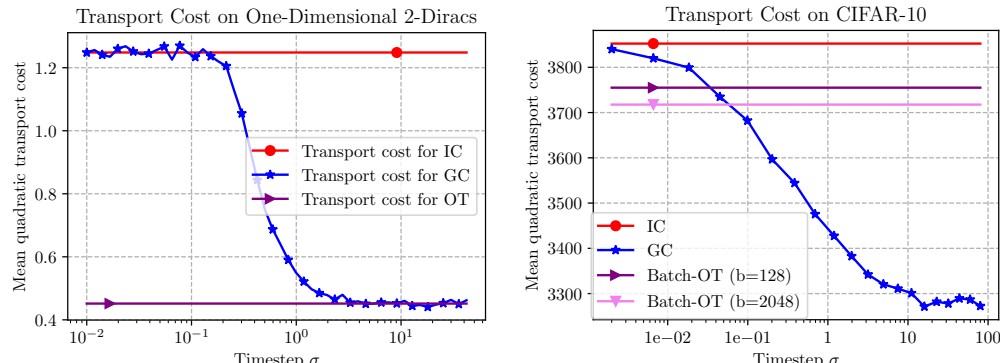

Figure 3: Comparison of transport costs between IC, batch-OT and GC on two datasets: a synthetic one presented in Figure 1 where $\mathbf{x}_\star \sim \{\delta_0, \delta_1\}$ (left) and CIFAR-10 (right).

## 5 IMAGE GENERATION WITH GENERATOR-INDUCED FLOWS

In this section, we present a careful analysis of training consistency models with GC on unconditional image generation. We identify two main obstacles to using GC: *(i)* the distribution shift in marginals between IC and GC affects the performance of consistency models trained with GC only, since $\hat{\mathbf{x}}_t$ is computed based on IC inputs; *(ii)* training a GC model with a consistency model pre-trained on IC is computationally costly. We circumvent these issues with a simple solution based on mixing IC and GC. Finally, we demonstrate that our mixing approach improves performance and accelerates convergence of consistency models.

Our experiments are done on the following datasets: CIFAR-10 (Krizhevsky, 2009), ImageNet (Deng et al., 2009), CelebA (Liu et al., 2015) and LSUN Church (Yu et al., 2015). For the evaluation metrics, we report the Fréchet Inception Distance (FID, Heusel et al. (2017)), Kernel Inception Distance (KID, Bińkowski et al. (2018)), and Inception Score (IS, Salimans et al. (2016)). Our experiments are based on the improved training techniques for consistency models from Song and Dhariwal (2024), denoted iCT-IC. Note that the iCT-IC reported below does not reach the same performance than in Song and Dhariwal (2024) since we need to use smaller model size and batch size. In our setting, on CIFAR-10, training iCT-IC requires approximately one day on a A100 40GB GPU. Details are provided in Appendix C, and the code is shared in the supplementary material for reproducibility.

### 5.1 GC FROM SCRATCH: THE CURSE OF DISTRIBUTION SHIFT

The first experiment involves training a consistency model with GC from scratch. As shown in Figure 4(a), we observe that these models converge quickly but reach saturation early in the training process. When applying the timestep scheduling method with an increasing number of timesteps from Song and Dhariwal (2024), the FID of the models worsens. Using a fixed number of timesteps prevents divergence of the FID, but it still plateaus at a higher FID than the baseline iCT-IC.

**Finding 1.** *Consistency models converge faster with GC compared to the standard IC approach.*

In Figure 4(b), we plot the FID per timestep for three model/trajectory pairs: GC-model on IC trajectories, GC-model on GC trajectories, and IC-model on IC trajectories. Notably, we observe the impact of the distribution shift between IC and GC trajectories: the FID of the GC-model on IC trajectories degrades at the intermediate timesteps of the diffusion process. This highlights why training a model exclusively on GC trajectories is insufficient: to build $\mathbf{x}_{t_i}$ in Equation (13), the model is inferred on IC but trained on GC trajectories. If IC and GC differ too significantly, the model cannot improve on IC.

**Finding 2.** *Consistency models trained exclusively on GC trajectories exhibit sub-optimal performance, likely due to a distribution shift between intermediate timesteps of IC and GC trajectories.*

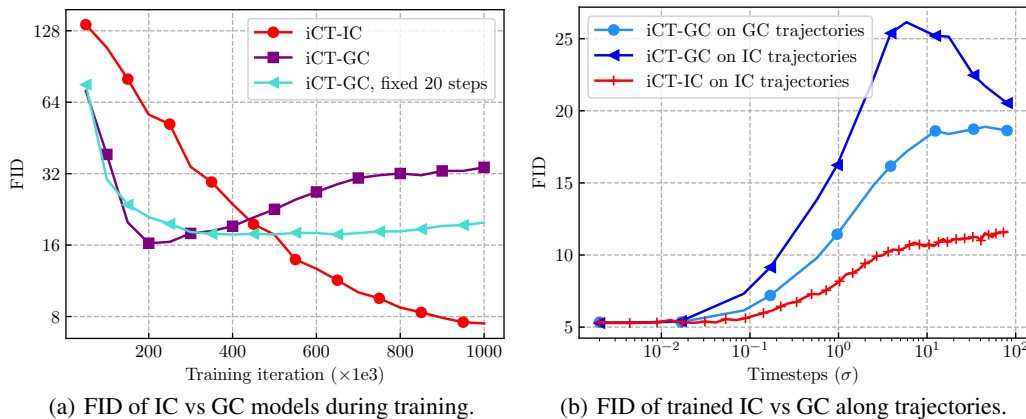

(a) FID of IC vs GC models during training.  (b) FID of trained IC vs GC along trajectories.

Figure 4: Analysis of GC model on CIFAR-10. (a) When trained with only GC trajectories, consistency models does not reach the performance of the base model (iCT-IC). In (b), we show that is linked to a distribution shift problem: GC models are weak on IC trajectoires, thus are sub-optimal for predicting $\hat{\mathbf{x}}_{t_i}$ required in their own training (Equation (13)).

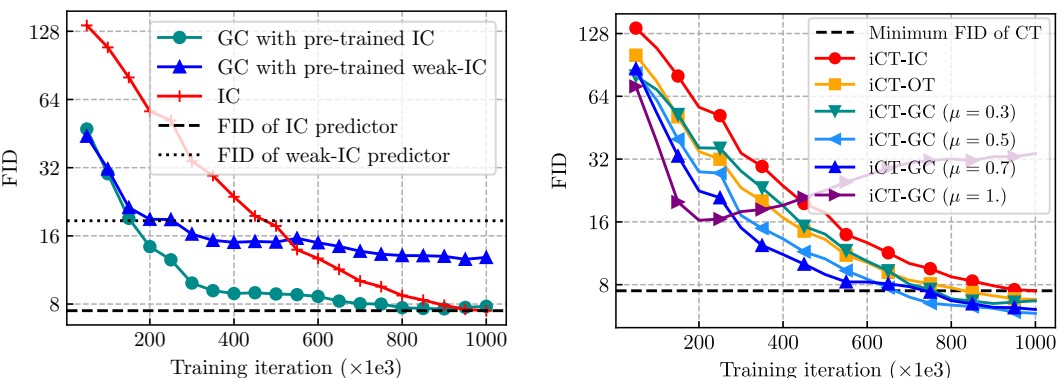

Figure 5: A pre-trained endpoint predictor for GC solves divergence issue observed previously on CIFAR-10. However, the performance of GC depends on the performance of the predictor.

Figure 6: Consistency models trained with GC with mixing converges faster and outperforms consistency models trained with IC or minibatch-OT on CIFAR-10.

## 5.2 GC WITH PRE-TRAINED ENDPOINT PREDICTOR

As a proof of concept and to further corroborate our intuition on Finding 2, we train a consistency model on GC with a separate endpoint predictor $\mathbf{g}_\phi$, pre-trained on IC (iCT-IC) and kept frozen: $\hat{\mathbf{x}}_{t_i} = \mathbf{g}_\phi(\mathbf{x}_{t_i}, \sigma_{t_i})$. In Figure 5, we report the performance of consistency models on CIFAR-10 trained with GC using two different $\mathbf{g}_\phi$: *(i)* a $\mathbf{g}_\phi$ fully-trained as standard iCT-IC with 100k training steps; *(ii)* a weak $\mathbf{g}_\phi$ partially-trained as iCT-IC with 20k training steps.

**Finding 3.** *Using a partially pre-trained and frozen endpoint predictor, trained on IC trajectories, prevents the consistency model trained on GC from worsening when increasing the number of timesteps. However, the performance of the GC model depends on the quality of the endpoint predictor evaluated on IC trajectories.*

It is important to note that this setup is not ideal, as it requires pre-training a standard consistency model. In practice, we aim for a training methodology that accelerates convergence and improves performance when training from scratch, without doubling the number of required training iterations.

Table 1: iCT-IC is the standard improved consistency model with independent coupling (Song and Dhariwal, 2024); iCT-OT is iCT with minibatch optimal transport coupling (Pooladian et al., 2023; Dou et al., 2024); iCT-GC ($\mu = 0.5$) is our proposed GC with mixing.

| Dataset | Model | FID $\downarrow$ | KID ($\times 10^2$) $\downarrow$ | IS $\uparrow$ |
|---|---|---|---|---|
| CIFAR-10 | iCT-IC | $7.42 \pm 0.04$ | $0.44 \pm 0.03$ | $8.76 \pm 0.06$ |
| | iCT-OT | $6.75 \pm 0.04$ | $0.36 \pm 0.04$ | $8.86 \pm 0.09$ |
| | iCT-GC ($\mu = 0.5$) | $\mathbf{5.95} \pm 0.05$ | $\mathbf{0.26} \pm 0.02$ | $\mathbf{9.10} \pm 0.05$ |
| ImageNet ($32 \times 32$) | iCT-IC | $14.89 \pm 0.17$ | $1.23 \pm 0.05$ | $9.46 \pm 0.06$ |
| | iCT-OT | $14.13 \pm 0.17$ | $1.18 \pm 0.05$ | $9.62 \pm 0.06$ |
| | iCT-GC ($\mu = 0.5$) | $\mathbf{13.99} \pm 0.28$ | $\mathbf{1.13} \pm 0.03$ | $\mathbf{9.77} \pm 0.07$ |
| CelebA ($64 \times 64$) | iCT-IC | $15.82 \pm 0.13$ | $1.31 \pm 0.04$ | $2.33 \pm 0.00$ |
| | iCT-OT | $13.63 \pm 0.13$ | $1.09 \pm 0.03$ | $2.40 \pm 0.01$ |
| | iCT-GC ($\mu = 0.5$) | $\mathbf{11.74} \pm 0.08$ | $\mathbf{0.91} \pm 0.04$ | $\mathbf{2.45} \pm 0.01$ |
| LSUN Church ($64 \times 64$) | iCT-IC | $10.58 \pm 0.11$ | $0.73 \pm 0.03$ | $1.99 \pm 0.01$ |
| | iCT-OT | $\mathbf{9.71} \pm 0.13$ | $\mathbf{0.64} \pm 0.03$ | $2.00 \pm 0.01$ |
| | iCT-GC ($\mu = 0.5$) | $9.88 \pm 0.07$ | $0.66 \pm 0.04$ | $\mathbf{2.14} \pm 0.01$ |

## 5.3 GC FROM SCRATCH WITH MIXING

We propose a simple yet effective solution to address the distribution shift issue that affects the performance and robustness of the endpoint predictor on IC trajectories: mixing IC and GC during training. We introduce a mixing factor $\mu$: at each training step, training pairs are drawn from GC with probability $\mu$, while the remaining pairs are drawn from standard IC. We denote this mixing procedure as GC ($\mu = \cdot$). Hence, GC ($\mu = 0$) corresponds to the standard IC procedure, while GC ($\mu = 1$) corresponds to the procedure introduced in Section 4. We apply this mixing approach to four image datasets, and include comparisons to batch OT (Pooladian et al., 2023; Dou et al., 2024) as an additional baseline. Results across multiple datasets and metrics are presented in Table 1, and visual examples are shown in Appendix Figure 7.

**Finding 4.** *Mixing IC and GC trajectories consistently improves results compared to the base IC model and outperforms batch-OT in most cases.*

As shown in Figure 6, we observe an interesting interpolation phenomenon between $\mu = 0$ and $\mu = 1$. At $\mu = 0$, we recover the steady FID improvement typical of IC training. As $\mu$ increases, the convergence of the generative model accelerates. At $\mu = 1$, we observe the fast convergence and early divergence described in Section 5.1. For $0.5 \leq \mu \leq 0.7$, we find a sweet spot where convergence speed and final FID are improved compared to IC and batch-OT models. We provide further detail on the sensitivity of our results to the choice of $\mu$, which we found easy to tune, in Appendices B.1 and C.

## 6 CONCLUSION

In this paper, we identify a discrepancy between consistency training and consistency distillation. Building on this theoretical analysis, we introduce generator-induced flows and show that they reduce a proxy term measuring this discrepancy. Additionally, generator-induced flows decrease the data-to-noise transport cost, as demonstrated by theory and experiments. Finally, we derive practical algorithms for training consistency models using generator-induced flows and demonstrate their improved empirical performance.

**Reproducibility Statement.** In this work, we are committed to maintaining high standards of reproducibility. On the theoretical side, all assumptions and proofs of our results can be found in Appendix A. For the experimental part, our codebase is included in the Supplementary Material and will be publicly released upon publication. Additionally, detailed information about datasets, architectures, and hyperparameters is provided in Appendix C.

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

# A PROOFS

## A.1 CONTINUOUS-TIME CONSISTENCY OBJECTIVES

**Theorem 1** (**Discrepancy between consistency distillation and consistency training objectives**).
Assume that the distance function is given by $\mathcal{D}(\mathbf{x}, \mathbf{y}) = \varphi(\|\mathbf{x} - \mathbf{y}\|)$ for a continuous convex
function $\varphi : [0, \infty) \to [0, \infty)$ with $\varphi(x) \sim Cx^\alpha$ as $x \to 0^+$ for some $C > 0$ and $\alpha \geq 1$, and that
the timesteps are equally spaced, i.e., $t_i = \frac{iT}{N}$. Furthermore, assume that the Jacobian $\frac{\partial \boldsymbol{f}_\theta}{\partial \mathbf{x}}$ does not
vanish identically. Then the following assertions hold:

(i) The scaled consistency losses $N^\alpha \mathcal{L}_{\mathrm{CD}}(\theta)$ and $N^\alpha \mathcal{L}_{\mathrm{CT}}(\theta)$ converge as $N \to \infty$. Moreover,
the minimization objectives corresponding to these limiting scaled consistency losses are
not equivalent, and their difference is given by:

$$\lim_{N \to \infty} N^\alpha \left[\mathcal{L}_{\mathrm{CT}}(\theta) - \mathcal{L}_{\mathrm{CD}}(\theta)\right] = CT^{\alpha-1}\mathcal{R}(\theta), \tag{22}$$

where $\mathcal{R}(\theta)$ is defined by

$$\mathcal{R}(\theta) = \int_0^T \lambda(\sigma_t)\mathbb{E}\left[\|\partial_{\mathrm{CT}}\boldsymbol{f}_\theta\|^\alpha - \|\partial_{\mathrm{CD}}\boldsymbol{f}_\theta\|^\alpha\right]\,\mathrm{d}t \tag{23}$$

and satisfies $\mathcal{R}(\theta) > 0$, with

$$\partial_{\mathrm{CT}}\boldsymbol{f}_\theta = \frac{\partial \boldsymbol{f}_\theta}{\partial \sigma}(\mathbf{x}_t, \sigma_t)\dot{\sigma}_t + \frac{\partial \boldsymbol{f}_\theta}{\partial \mathbf{x}}(\mathbf{x}_t, \sigma_t) \cdot \dot{\mathbf{x}}_t, \tag{24}$$

$$\partial_{\mathrm{CD}}\boldsymbol{f}_\theta = \frac{\partial \boldsymbol{f}_\theta}{\partial \sigma}(\mathbf{x}_t, \sigma_t)\dot{\sigma}_t + \frac{\partial \boldsymbol{f}_\theta}{\partial \mathbf{x}}(\mathbf{x}_t, \sigma_t) \cdot \mathbf{v}_t(\mathbf{x}_t). \tag{25}$$

In particular, if $\alpha = 2$,

$$\mathcal{R}(\theta) = \int_0^T \lambda(\sigma_t)\mathbb{E}\left[\left\|\frac{\partial \boldsymbol{f}_\theta}{\partial \mathbf{x}}(\mathbf{x}_t, \sigma_t) \cdot \left(\dot{\mathbf{x}}_t - \mathbf{v}_t(\mathbf{x}_t)\right)\right\|^2\right]\,\mathrm{d}t. \tag{26}$$

(ii) The scaled gradient $N^{\alpha-1}\nabla_\theta \mathcal{L}_{\mathrm{CD}}(\theta)$ and $N^{\alpha-1}\nabla_\theta \mathcal{L}_{\mathrm{CT}}(\theta)$ converge as $N \to \infty$. More-
over, if $\alpha \neq 2$, then their respective limits are not identical as functions of $\theta$:

$$\lim_{N \to \infty} N^{\alpha-1}\nabla_\theta \mathcal{L}_{\mathrm{CT}}(\theta) \neq \lim_{N \to \infty} N^{\alpha-1}\nabla_\theta \mathcal{L}_{\mathrm{CD}}(\theta). \tag{27}$$

*Proof.* *(i)* Note that $\partial_{\mathrm{CD}}\boldsymbol{f}_\theta$ and $\partial_{\mathrm{CT}}\boldsymbol{f}_\theta$ satisfy:

$$\partial_{\mathrm{CT}}\boldsymbol{f}_\theta(\mathbf{x}_t, \sigma_t) = \frac{\partial}{\partial t}\boldsymbol{f}_\theta(\mathbf{x}_t, \sigma_t), \qquad \partial_{\mathrm{CD}}\boldsymbol{f}_\theta(\mathbf{x}_t, \sigma_t) = \mathbb{E}\left[\frac{\partial}{\partial t}\boldsymbol{f}_\theta(\mathbf{x}_t, \sigma_t)\,\Big|\,\mathbf{x}_t\right]. \tag{28}$$

Here, the second equality follows by noting that $\mathbf{v}_t(\mathbf{x}_t) = \mathbb{E}[\dot{\mathbf{x}}_t|\mathbf{x}_t]$ and all the other terms in the
expansion of $\frac{\partial}{\partial t}\boldsymbol{f}_\theta(\mathbf{x}_t, \sigma_t)$ are completely determined once the value of $\mathbf{x}_t$ is known.

Now, we use Taylor's theorem to expand the difference between $\boldsymbol{f}_\theta(\mathbf{x}_{t_{i+1}}, \sigma_{t_{i+1}})$ and $\boldsymbol{f}_\theta(\mathbf{x}_{t_i}^\Phi, \sigma_{t_i})$ in
the consistency distillation loss, Equation (4). Together with the definition of $\mathbf{x}_{t_i}^\Phi$, Equation (5), this
yields:

$$\boldsymbol{f}_\theta(\mathbf{x}_{t_{i+1}}, \sigma_{t_{i+1}}) - \boldsymbol{f}_\theta(\mathbf{x}_{t_i}^\Phi, \sigma_{t_i})$$
$$= \frac{\partial \boldsymbol{f}_\theta}{\partial \sigma}(\mathbf{x}_{t_{i+1}}, \sigma_{t_{i+1}}) \cdot (\sigma_{t_{i+1}} - \sigma_{t_i}) + \frac{\partial \boldsymbol{f}_\theta}{\partial \mathbf{x}}(\mathbf{x}_{t_{i+1}}, \sigma_{t_{i+1}}) \cdot (\mathbf{x}_{t_{i+1}} - \mathbf{x}_{t_i}^\Phi) + o(t_{i+1} - t_i) \tag{29}$$
$$= \partial_{\mathrm{CD}}\boldsymbol{f}_\theta(\mathbf{x}_{t_{i+1}}, \sigma_{t_{i+1}}) \cdot (t_{i+1} - t_i) + o(t_{i+1} - t_i). \tag{30}$$

Similarly, by expanding the difference between $\boldsymbol{f}_\theta(\mathbf{x}_{t_{i+1}}, \sigma_{t_{i+1}})$ and $\boldsymbol{f}_\theta(\mathbf{x}_{t_i}, \sigma_{t_i})$ in Equation (6),

$$\boldsymbol{f}_\theta(\mathbf{x}_{t_{i+1}}, \sigma_{t_{i+1}}) - \boldsymbol{f}_\theta(\mathbf{x}_{t_i}, \sigma_{t_i})$$
$$= \frac{\partial \boldsymbol{f}_\theta}{\partial \sigma}(\mathbf{x}_{t_{i+1}}, \sigma_{t_{i+1}}) \cdot (\sigma_{t_{i+1}} - \sigma_{t_i}) + \frac{\partial \boldsymbol{f}_\theta}{\partial \mathbf{x}}(\mathbf{x}_{t_{i+1}}, \sigma_{t_{i+1}}) \cdot (\mathbf{x}_{t_{i+1}} - \mathbf{x}_{t_i}) + o(t_{i+1} - t_i) \tag{31}$$
$$= \partial_{\mathrm{CT}}\boldsymbol{f}_\theta(\mathbf{x}_{t_{i+1}}, \sigma_{t_{i+1}}) \cdot (t_{i+1} - t_i) + o(t_{i+1} - t_i). \tag{32}$$

Therefore, for each $\bullet \in \{\mathrm{CD}, \mathrm{CT}\}$,

$$N^\alpha \mathcal{L}_\bullet(\theta) = N^\alpha \cdot \frac{1}{N} \sum_{i=0}^{N-1} \lambda(\sigma_{t_i}) \mathbb{E}\left[ C \left\| \partial_\bullet \boldsymbol{f}_\theta(\mathbf{x}_{t_{i+1}}, \sigma_{t_{i+1}}) \right\|^\alpha (1 + o(1)) \right] \cdot (t_{i+1} - t_i)^\alpha \quad (33)$$

$$= CT^{\alpha-1} \sum_{i=0}^{N-1} \lambda(\sigma_{t_i}) \mathbb{E}\left[ \left\| \partial_\bullet \boldsymbol{f}_\theta(\mathbf{x}_{t_{i+1}}, \sigma_{t_{i+1}}) \right\|^\alpha (1 + o(1)) \right] \cdot (t_{i+1} - t_i) \quad (34)$$

$$\to CT^{\alpha-1} \int_0^T \lambda(\sigma_t) \mathbb{E}\left[ \left\| \partial_\bullet \boldsymbol{f}_\theta(\mathbf{x}_t, \sigma_t) \right\|^\alpha \right] \mathrm{d}t \quad (35)$$

in the continuous-time limit as $N \to \infty$.

For simplicity of notation, we write

$$\mathcal{L}_\bullet^\infty(\theta) = \lim_{N \to \infty} N^\alpha \mathcal{L}_\bullet(\theta) \quad (36)$$

for each $\bullet \in \{\mathrm{CD}, \mathrm{CT}\}$. Then, from the formula for the limiting losses $\mathcal{L}_\bullet^\infty(\theta)$, Equation (35), we immediately obtain

$$\mathcal{L}_{\mathrm{CT}}^\infty(\theta) - \mathcal{L}_{\mathrm{CD}}^\infty(\theta) = CT^{\alpha-1} \int_0^T \lambda(\sigma_t) \mathbb{E}\left[ \left\| \partial_{\mathrm{CT}} \boldsymbol{f}_\theta(\mathbf{x}_t, \sigma_t) \right\|^\alpha - \left\| \partial_{\mathrm{CD}} \boldsymbol{f}_\theta(\mathbf{x}_t, \sigma_t) \right\|^\alpha \right] \mathrm{d}t. \quad (37)$$

Now, we specialize in the case $\alpha = 2$ and invoke the general observation that, for any random vectors $\mathbf{x}$ and $\mathbf{y}$, the following identity holds:

$$\mathbb{E}\left[ \|\mathbf{x}\|^2 - \|\mathbb{E}[\mathbf{x}|\mathbf{y}]\|^2 \right] = \mathbb{E}\left[ \|\mathbf{x} - \mathbb{E}[\mathbf{x}|\mathbf{y}]\|^2 \right]. \quad (38)$$

This can be easily proved by expanding the squared Euclidean norm as the inner product and applying the law of iterated expectations. Plugging in $\mathbf{x} \leftarrow \frac{\partial}{\partial t} \boldsymbol{f}_\theta(\mathbf{x}_t, \sigma_t)$ and $\mathbf{y} \leftarrow \mathbf{x}_t$, and noting that $\partial_{\mathrm{CD}} \boldsymbol{f}_\theta(\mathbf{x}_t, \sigma_t) = \mathbb{E}\left[ \partial_{\mathrm{CT}} \boldsymbol{f}_\theta(\mathbf{x}_t, \sigma_t) \mid \mathbf{x}_t \right]$ by Equation (28), it follows that

$$\mathcal{L}_{\mathrm{CT}}^\infty(\theta) - \mathcal{L}_{\mathrm{CD}}^\infty(\theta) = CT \int_0^T \lambda(\sigma_t) \mathbb{E}\left[ \left\| \partial_{\mathrm{CT}} \boldsymbol{f}_\theta(\mathbf{x}_t, \sigma_t) - \partial_{\mathrm{CD}} \boldsymbol{f}_\theta(\mathbf{x}_t, \sigma_t) \right\|^2 \right] \mathrm{d}t \quad (39)$$

$$= CT \int_0^T \lambda(\sigma_t) \mathbb{E}\left[ \left\| \frac{\partial \boldsymbol{f}_\theta}{\partial \mathbf{x}}(\mathbf{x}_t, \sigma_t) \cdot (\dot{\mathbf{x}}_t - \mathbf{v}_t(\mathbf{x}_t)) \right\|^2 \right] \mathrm{d}t. \quad (40)$$

Next, we establish the positivity of $\mathcal{R}(\theta)$. To this end, note that $\| \cdot \|^\alpha$ is a convex function for $\alpha \geq 1$. By invoking the conditional Jensen's inequality, we find that the expectation inside the limiting scaled consistency training losses, Equation (35) satisfy:

$$\mathbb{E}\left[ \left\| \partial_{\mathrm{CT}} \boldsymbol{f}_\theta(\mathbf{x}_t, \sigma_t) \right\|^\alpha \right] = \mathbb{E}\left[ \left\| \frac{\partial}{\partial t} \boldsymbol{f}_\theta(\mathbf{x}_t, \sigma_t) \right\|^\alpha \right] = \mathbb{E}\left[ \mathbb{E}\left[ \left\| \frac{\partial}{\partial t} \boldsymbol{f}_\theta(\mathbf{x}_t, \sigma_t) \right\|^\alpha \;\middle|\; \mathbf{x}_t \right] \right] \quad (41)$$

$$\geq \mathbb{E}\left[ \left\| \mathbb{E}\left[ \frac{\partial}{\partial t} \boldsymbol{f}_\theta(\mathbf{x}_t, \sigma_t) \;\middle|\; \mathbf{x}_t \right] \right\|^\alpha \right] = \mathbb{E}\left[ \left\| \partial_{\mathrm{CD}} \boldsymbol{f}_\theta(\mathbf{x}_t, \sigma_t) \right\|^\alpha \right]. \quad (42)$$

Integrating both sides with respect to $\lambda(\sigma_t) \, \mathrm{d}t$, we obtain the desired inequality. The Jensen's inequality also tells that the equality holds precisely when $\frac{\partial}{\partial t} \boldsymbol{f}_\theta(\mathbf{x}_t, \sigma_t) = \mathbb{E}[\frac{\partial}{\partial t} \boldsymbol{f}_\theta(\mathbf{x}_t, \sigma_t)|\mathbf{x}_t]$ holds, or equivalently, $\frac{\partial \boldsymbol{f}_\theta}{\partial \mathbf{x}}(\mathbf{x}_t, \sigma_t) \cdot (\dot{\mathbf{x}}_t - \mathbb{E}[\dot{\mathbf{x}}_t|\mathbf{x}_t]) = 0$. However, given the value of $\mathbf{x}_t$, the quantity $\dot{\mathbf{x}}_t$ can assume an arbitrary value in $\mathbb{R}^d$ because the conditional density of $\dot{\mathbf{x}}_t = \dot{\sigma}_t \mathbf{z}$ given $\mathbf{x}_t$ is strictly positive everywhere. Consequently, the equality condition implies $\frac{\partial \boldsymbol{f}_\theta}{\partial \mathbf{x}} = 0$. Since this contradicts the assumption of the theorem, the strict inequality between the two limiting losses must hold.

Finally, recall that the continuous-time consistency distillation loss, $\mathcal{L}_{\mathrm{CD}}^\infty(\theta)$, is given by

$$\mathcal{L}_{\mathrm{CD}}^\infty(\theta) = CT^{\alpha-1} \int_0^T \lambda(\sigma_t) \mathbb{E}\left[ \left\| \frac{\partial \boldsymbol{f}_\theta}{\partial \sigma}(\mathbf{x}_t, \sigma_t) \dot{\sigma}_t + \frac{\partial \boldsymbol{f}_\theta}{\partial \mathbf{x}}(\mathbf{x}_t, \sigma_t) \cdot \mathbf{v}_t(\mathbf{x}_t) \right\|^\alpha \right] \mathrm{d}t. \quad (43)$$

Similarly, the continuous-time consistency training loss, $\mathcal{L}_{\text{CT}}^{\infty}(\theta)$, is given by

$$\mathcal{L}_{\text{CT}}^{\infty}(\theta) = CT^{\alpha-1} \int_0^T \lambda(\sigma_t) \mathbb{E}\left[\left\|\frac{\partial \boldsymbol{f}_\theta}{\partial \sigma}(\mathbf{x}_t, \sigma_t)\dot{\sigma}_t + \frac{\partial \boldsymbol{f}_\theta}{\partial \mathbf{x}}(\mathbf{x}_t, \sigma_t) \cdot \dot{\mathbf{x}}_t\right\|^{\alpha}\right] \mathrm{d}t. \tag{44}$$

Since $\mathbf{v}_t(\mathbf{x}_t) = \mathbb{E}[\dot{\mathbf{x}}_t | \mathbf{x}_t]$ and $\mathbb{E}[\|\dot{\mathbf{x}}_t - \mathbb{E}[\dot{\mathbf{x}}_t]\|^2] > \mathbb{E}[\|\mathbf{v}_t(\mathbf{x}_t) - \mathbb{E}[\dot{\mathbf{x}}_t]\|^2]$, it follows that $\mathcal{L}_{\text{CT}}^{\infty}(\theta)$ penalizes the Jacobian $\frac{\partial \boldsymbol{f}_\theta}{\partial \mathbf{x}}$ more strongly than $\mathcal{L}_{\text{CD}}^{\infty}(\theta)$ does. Therefore, the two limiting consistency losses do not define equivalent objectives.

*(ii)* Using the convexity of $\varphi$, we can show that $\varphi'(x) \sim C\alpha x^{\alpha-1}$ as $x \to 0^+$. Combining this with the vector calculus formula $\nabla_{\mathbf{y}}\|\mathbf{y}\| = \frac{\mathbf{y}}{\|\mathbf{y}\|}$, we get $\nabla_{\mathbf{y}}\varphi(\|\mathbf{y}\|) \approx C\alpha\|\mathbf{y}\|^{\alpha-2}\mathbf{y}$ for small $\mathbf{y}$. From this, we can estimate the gradient of the distance between $\text{sg}(\boldsymbol{f}_\theta(\mathbf{x}_{t_i}^{\Phi}, \sigma_{t_i}))$ and $\boldsymbol{f}_\theta(\mathbf{x}_{t_{i+1}}, \sigma_{t_{i+1}})$ with respect to the model parameter $\theta$ as:

$$\nabla_\theta \mathcal{D}\big(\text{sg}\big(\boldsymbol{f}_\theta(\mathbf{x}_{t_i}^{\Phi}, \sigma_{t_i})\big), \boldsymbol{f}_\theta(\mathbf{x}_{t_{i+1}}, \sigma_{t_{i+1}})\big)$$
$$= (1 + o(1))C\alpha \left[\|\partial_{\text{CD}}\boldsymbol{f}_\theta\|^{\alpha-2}(\partial_{\text{CD}}\boldsymbol{f}_\theta)^\top \frac{\partial \boldsymbol{f}_\theta}{\partial \theta}\right] \cdot (t_{i+1} - t_i)^{\alpha-1} \tag{45}$$

Here, the expression $\|\partial_{\text{CD}}\boldsymbol{f}_\theta\|^{\alpha-2}(\partial_{\text{CD}}\boldsymbol{f}_\theta)^\top \frac{\partial \boldsymbol{f}_\theta}{\partial \theta}$ in the square bracket is evaluated at $(\mathbf{x}_{t_{i+1}}, \sigma_{t_{i+1}})$. Similarly, the gradient of the distance between $\boldsymbol{f}_\theta(\mathbf{x}_{t_i}, \sigma_{t_i})$ and $\boldsymbol{f}_\theta(\mathbf{x}_{t_{i+1}}, \sigma_{t_{i+1}})$ is estimated as:

$$\nabla_\theta \mathcal{D}\big(\text{sg}\big(\boldsymbol{f}_\theta(\mathbf{x}_{t_i}, \sigma_{t_i})\big), \boldsymbol{f}_\theta(\mathbf{x}_{t_{i+1}}, \sigma_{t_{i+1}})\big)$$
$$= (1 + o(1))C\alpha \left[\|\partial_{\text{CT}}\boldsymbol{f}_\theta\|^{\alpha-2}(\partial_{\text{CT}}\boldsymbol{f}_\theta)^\top \frac{\partial \boldsymbol{f}_\theta}{\partial \theta}\right] \cdot (t_{i+1} - t_i)^{\alpha-1} \tag{46}$$

Combining these two estimates, we can now compute the limit of the scaled gradient $N^{\alpha-1}\nabla_\theta \mathcal{L}_\bullet(\theta)$ for each $\bullet \in \{\text{CD}, \text{CT}\}$ as:

$$N^{\alpha-1}\nabla_\theta \mathcal{L}_\bullet(\theta)$$
$$= C\alpha T^{\alpha-2} \sum_{i=0}^{N-1} \lambda(\sigma_{t_i})\mathbb{E}\left[(1 + o(1))\left[\|\partial_\bullet \boldsymbol{f}_\theta\|^{\alpha-2}(\partial_\bullet \boldsymbol{f}_\theta)^\top \frac{\partial \boldsymbol{f}_\theta}{\partial \theta}\right]\right] \cdot (t_{i+1} - t_i) \tag{47}$$

$$\to C\alpha T^{\alpha-2} \int_0^T \lambda(\sigma_t)\mathbb{E}\left[\|\partial_\bullet \boldsymbol{f}_\theta(\mathbf{x}_t, \sigma_t)\|^{\alpha-2}(\partial_\bullet \boldsymbol{f}_\theta(\mathbf{x}_t, \sigma_t))^\top \frac{\partial \boldsymbol{f}_\theta}{\partial \theta}(\mathbf{x}_t, \sigma_t)\right] \mathrm{d}t \tag{48}$$

as $N \to \infty$. Finally, if $\alpha \neq 2$, then the term $\|\partial_\bullet \boldsymbol{f}_\theta\|^{\alpha-2} \partial_\bullet \boldsymbol{f}_\theta^\top$ is a nonlinear transformation of $\partial_\bullet \boldsymbol{f}_\theta$. This nonlinearity tells that, in general,

$$\mathbb{E}\left[\|\partial_{\text{CT}}\boldsymbol{f}_\theta\|^{\alpha-2}(\partial_{\text{CT}}\boldsymbol{f}_\theta)^\top \Big| \mathbf{x}_t\right] \neq \|\partial_{\text{CD}}\boldsymbol{f}_\theta\|^{\alpha-2}(\partial_{\text{CD}}\boldsymbol{f}_\theta)^\top. \tag{49}$$

Therefore, the scaled gradient limits are not identical as functions of $\theta$, and in particular, their zero sets do not coincide. $\qquad\square$

### A.2 PROXY OF THE REGULARIZER

In this subsection, we establish a theoretical result about the decay rate of the proxy of the regularizer. As preparation for the main result and for future use, we introduce a simple lemma that decomposes the forward flow generated by a vector field into the sum of a scaling term and a correction term that is well-behaved.

**Lemma 2.** *Assume that $\phi$ is the forward flow generated by the vector field $\mathbf{v}_t$, meaning that it solves the characteristic equation:*

$$\frac{\partial}{\partial t}\phi(\mathbf{x}, \sigma_t) = \mathbf{v}_t(\phi(\mathbf{x}, \sigma_t)), \qquad \phi(\mathbf{x}, \sigma_0) = \mathbf{x}. \tag{50}$$

*Also, assume that $\mathbf{v}_t$ is defined as*

$$\mathbf{v}_t(\mathbf{x}) = \frac{\dot{\sigma}_t}{\sigma_t}(\mathbf{x} - \boldsymbol{D}(\mathbf{x}, \sigma_t)) \tag{51}$$

*for some function $\boldsymbol{D}$, which we call a "denoiser". Then $\phi$ satisfies the following integral equation:*

$$\mathbf{x} = \frac{\sigma_0}{\sigma_t}\phi(\mathbf{x}, \sigma_t) + \sigma_0 \int_0^t \frac{\dot{\sigma}_s}{\sigma_s^2}\boldsymbol{D}(\phi(\mathbf{x}, \sigma_s), \sigma_s) \, \mathrm{d}s. \tag{52}$$

*Proof.* We first compute the derivative of $\phi/\sigma_t$:

$$\frac{\partial}{\partial t}\left(\frac{\phi(\mathbf{x},\sigma_t)}{\sigma_t}\right) = -\frac{\dot{\sigma}_t}{\sigma_t^2}\phi(\mathbf{x},\sigma_t) + \frac{1}{\sigma_t}\cdot\frac{\dot{\sigma}_t}{\sigma_t}(\phi(\mathbf{x},\sigma_t) - \boldsymbol{D}(\phi(\mathbf{x},\sigma_t),\sigma_t)) \tag{53}$$

$$= -\frac{\dot{\sigma}_t}{\sigma_t^2}\boldsymbol{D}(\phi(\mathbf{x},\sigma_t),\sigma_t). \tag{54}$$

Integrating both sides with respect to $t$, it follows that

$$\frac{\phi(\mathbf{x},\sigma_t)}{\sigma_t} - \frac{\phi(\mathbf{x},\sigma_0)}{\sigma_0} = -\int_0^t \frac{\dot{\sigma}_s}{\sigma_s^2}\boldsymbol{D}(\phi(\mathbf{x},\sigma_s),\sigma_s)\,\mathrm{d}s. \tag{55}$$

Rearranging and applying the initial condition $\phi(\mathbf{x},\sigma_0) = \mathbf{x}$, we obtain the desired equation. $\qquad\square$

As an immediate consequence of this lemma, we obtain the following result about the asymptotic structure of a trained consistency model:

**Lemma 3.** *Assume that $\mathring{\boldsymbol{f}}$ is the consistency model generated by a bounded denoiser $\boldsymbol{D}$, in the sense that $\mathring{\boldsymbol{f}}$ solves the transport equation*

$$\frac{\partial\mathring{\boldsymbol{f}}}{\partial\sigma}(\mathbf{x},\sigma_t)\dot{\sigma}_t + \frac{\partial\mathring{\boldsymbol{f}}}{\partial\mathbf{x}}(\mathbf{x},\sigma_t)\cdot\mathbf{v}_t(\mathbf{x}) = 0 \tag{56}$$

*for a vector field $\mathring{\mathbf{v}}_t$ defined as in Equation (51) with the denoiser $\boldsymbol{D}$. Then*

$$\mathring{\boldsymbol{f}}(\mathbf{x},\sigma_t) = \frac{\sigma_0}{\sigma_t}\mathbf{x} + \mathcal{O}(1) \tag{57}$$

*uniformly in $\mathbf{x}$ and $\sigma_t$. The implicit bound of the error term can be chosen to be the bound of $\boldsymbol{D}$.*

*Proof.* Let $\phi$ be the forward flow generated by $\mathring{\mathbf{v}}_t$ as in Lemma 2. This $\phi$ is precisely the inverse of the consistency model $\mathring{\boldsymbol{f}}$, in the sense that $\phi(\mathring{\boldsymbol{f}}(\mathbf{x},\sigma),\sigma) = \mathbf{x}$ holds. Then, replacing $\mathbf{x}$ in the equation of Lemma 2 with $\mathring{\boldsymbol{f}}(\mathbf{x},\sigma_t)$, we get

$$\mathring{\boldsymbol{f}}(\mathbf{x},\sigma_t) = \frac{\sigma_0}{\sigma_t}\mathbf{x} + \sigma_0\int_0^t \frac{\dot{\sigma}_s}{\sigma_s^2}\boldsymbol{D}(\phi(\mathring{\boldsymbol{f}}(\mathbf{x},\sigma_t),\sigma_s),\sigma_s)\,\mathrm{d}s. \tag{58}$$

Now let $R$ be such that $\|\boldsymbol{D}(\mathbf{x},\sigma)\| \leq R$ for any $\mathbf{x}\in\mathbb{R}^d$ and noise level $\sigma$. Then, the integral term in Equation (58) is bounded as:

$$\left\|\sigma_0\int_0^t \frac{\dot{\sigma}_s}{\sigma_s^2}\boldsymbol{D}(\phi(\mathring{\boldsymbol{f}}(\mathbf{x},\sigma_t),\sigma_s),\sigma_s)\,\mathrm{d}s\right\| \leq \sigma_0\int_0^t \frac{\dot{\sigma}_s}{\sigma_s^2}R\,\mathrm{d}s = \sigma_0 R\left(\frac{1}{\sigma_0} - \frac{1}{\sigma_t}\right) \leq R. \tag{59}$$

This proves the desired claim. $\qquad\square$

Now we turn to the main result, which analyzes the asymptotic behavior of $\tilde{\mathcal{R}}_{t,\mathrm{IC}}$ and $\tilde{\mathcal{R}}_{t,\mathrm{GC}}$, as $t\to\infty$:

**Theorem 2.** Assume that the data distribution contains more than a single point. Also, assume that the generator-induced coupling between the predicted data point $\hat{\mathbf{x}}_t$ and noise $\mathbf{z}$ is computed via an ideal consistency model $\mathring{\boldsymbol{f}}$, *i.e.*, the flow of the PF-ODE. Then, as $t\to\infty$,

$$\tilde{\mathcal{R}}_{t,\mathrm{GC}} \ll \tilde{\mathcal{R}}_{t,\mathrm{IC}}. \tag{60}$$

*Proof.* We first investigate the asymptotic behavior of $\tilde{\mathcal{R}}_{t,\mathrm{IC}}$ in the limit of $t\to\infty$. Recall that the diffusion process $\mathbf{x}_t$ is given by $\mathbf{x}_t = \mathbf{x}_\star + \sigma_t\mathbf{z}$ for $(\mathbf{x}_\star,\mathbf{z})\sim q_\mathrm{I}$, and note that

$$\dot{\mathbf{x}}_t - \mathbf{v}_t(\mathbf{x}_t) = \dot{\sigma}_t\mathbf{z} - \mathbb{E}[\dot{\sigma}_t\mathbf{z}|\mathbf{x}_t] = -\frac{\dot{\sigma}_t}{\sigma_t}(\mathbf{x}_\star - \boldsymbol{D}(\mathbf{x}_t,\sigma_t)), \tag{61}$$

where $\boldsymbol{D}(\mathbf{x}_t,\sigma_t) = \mathbb{E}[\mathbf{x}_\star|\mathbf{x}_t]$ is the denoiser. Plugging this into the definition of $\tilde{\mathcal{R}}_{t,\mathrm{IC}}$, we get

$$\tilde{\mathcal{R}}_{t,\mathrm{IC}} = \left(\frac{\dot{\sigma}_t}{\sigma_t}\right)^2\mathbb{E}\left[\|\mathbf{x}_\star - \boldsymbol{D}(\mathbf{x}_t,\sigma_t)\|^2\right]. \tag{62}$$

Now, we claim that $D(\mathbf{x}_t, \sigma_t) = \mathbb{E}[\mathbf{x}_\star | \mathbf{x}_t] \to \mathbb{E}[\mathbf{x}_\star]$ as $t \to \infty$. Intuitively, this is because $\mathbf{x}_t \approx \sigma_t \mathbf{z}$ for large $t$, and $\sigma_t \mathbf{z}$ is independent of $\mathbf{x}_\star$. More formally, note that the conditional distribution of $\mathbf{x}_t$ given $\mathbf{x}_\star$ is $p(\mathbf{x}_t | \mathbf{x}_\star) = \mathcal{N}(\mathbf{x}_t; \mathbf{x}_\star, \sigma_t^2 \mathbf{I})$. By Bayes' theorem, the conditional distribution of $\mathbf{x}_\star$ given $\mathbf{x}_t$ is

$$p(\mathbf{x}_\star | \mathbf{x}_t) = \frac{p(\mathbf{x}_t | \mathbf{x}_\star) p(\mathbf{x}_\star)}{\int_{\mathbb{R}^d} p(\mathbf{x}_t | \mathbf{x}'_\star) p(\mathbf{x}'_\star) \, d\mathbf{x}'_\star} = \frac{\exp\left(-\frac{1}{2\sigma_t^2} |\mathbf{x}_t - \mathbf{x}_\star|^2\right) p(\mathbf{x}_\star)}{\int_{\mathbb{R}^d} \exp\left(-\frac{1}{2\sigma_t^2} |\mathbf{x}_t - \mathbf{x}'_\star|^2\right) p(\mathbf{x}'_\star) \, d\mathbf{x}'_\star}. \quad (63)$$

As $t \to \infty$, we have $\sigma_t \to \infty$, so the exponential terms converge to 1. Consequently, $p(\mathbf{x}_\star | \mathbf{x}_t) \to p(\mathbf{x}_\star)$ and hence $\mathbb{E}[\mathbf{x}_\star | \mathbf{x}_t] \to \mathbb{E}[\mathbf{x}_\star]$ as claimed. Thus,

$$\tilde{\mathcal{R}}_{t,\mathrm{IC}} \sim \left(\frac{\dot{\sigma}_t}{\sigma_t}\right)^2 \mathbb{E}\left[\left\|\mathbf{x}_\star - \mathbb{E}[\mathbf{x}_\star]\right\|^2\right]. \quad (64)$$

Since the data distribution $p_\star$ is assumed to have more than one point, the variance $\mathbb{E}[\|\mathbf{x}_\star - \mathbb{E}[\mathbf{x}_\star]\|^2]$ is strictly positive. Therefore, $\tilde{\mathcal{R}}_{t,\mathrm{IC}}$ decays at a rate asymptotically proportional to $(\frac{\dot{\sigma}_t}{\sigma_t})^2$.

Next, we investigate the asymptotic behavior of $\tilde{\mathcal{R}}_{t,\mathrm{GC}}$. Recall the consistency training loss for GC, Equation (15). Under the assumptions in Theorem 1, the scaled loss $N^\alpha \mathcal{L}_{\mathrm{GC}}(\theta)$ converges to

$$\mathcal{L}_{\mathrm{GC}}^\infty(\theta) = CT^{\alpha-1} \int_0^T \lambda(\sigma_t) \mathbb{E}\left[\left\|\frac{\partial \boldsymbol{f}_\theta}{\partial \sigma}(\tilde{\mathbf{x}}_t, \sigma_t) \dot{\sigma}_t + \frac{\partial \boldsymbol{f}_\theta}{\partial \mathbf{x}}(\tilde{\mathbf{x}}_t, \sigma_t) \cdot \dot{\sigma}_t \mathbf{z}\right\|^\alpha\right] dt. \quad (65)$$

Here, $\tilde{\mathbf{x}}_t = \hat{\mathbf{x}}_t + \sigma_t \mathbf{z}$ and $\hat{\mathbf{x}}_t = \mathring{\boldsymbol{f}}(\mathbf{x}_t, \sigma_t)$, where $\mathring{\boldsymbol{f}}$ is the ideal consistency model for the flow associated with the diffusion process $\mathbf{x}_t$. The proof of this claim is similar to that of Theorem 1, so we only highlight the necessary changes. Most importantly, the velocity term is not $\dot{\tilde{\mathbf{x}}}_t$ but $\dot{\sigma}_t \mathbf{z}$. This is due to how the discrete-time samples are constructed. Indeed, from Equation (14), we find that $\tilde{\mathbf{x}}_{t_{i+1}} - \tilde{\mathbf{x}}_{t_i} = (\sigma_{t_{i+1}} - \sigma_{t_i})\mathbf{z}$, which manifests as the velocity term $\dot{\sigma}_t \mathbf{z}$ in Equation (65). Consequently, the associated (average) velocity field $\tilde{\mathbf{v}}_t$ is given by

$$\tilde{\mathbf{v}}_t(\tilde{\mathbf{x}}_t) = \mathbb{E}[\dot{\sigma}_t \mathbf{z} | \tilde{\mathbf{x}}_t] = \frac{\dot{\sigma}_t}{\sigma_t}(\tilde{\mathbf{x}}_t - \mathbb{E}[\hat{\mathbf{x}}_t | \tilde{\mathbf{x}}_t]). \quad (66)$$

Therefore, $\tilde{\mathcal{R}}_{t,\mathrm{GC}}$ reduces to

$$\tilde{\mathcal{R}}_{t,\mathrm{GC}} = \left(\frac{\dot{\sigma}_t}{\sigma_t}\right)^2 \mathbb{E}\left[\left\|\hat{\mathbf{x}}_t - \mathbb{E}[\hat{\mathbf{x}}_t | \tilde{\mathbf{x}}_t]\right\|^2\right]. \quad (67)$$

Now, unlike in the IC case, we claim that $\mathbb{E}[\hat{\mathbf{x}}_t | \tilde{\mathbf{x}}_t] \approx \hat{\mathbf{x}}_t$ as $t \to \infty$. Heuristically, this is because both $\hat{\mathbf{x}}_t$ and $\tilde{\mathbf{x}}_t$ are almost deterministic functions of $\mathbf{z}$; hence, the conditioning has negligible effect in the limit.

More precisely, let $\phi$ be the forward flow generated by the PF-ODE vector field $\mathbf{v}_t$. As in the proof of Lemma 2, integrating both sides of Equation (54) from $t$ to $u$ yields

$$\frac{\phi(\mathbf{x}, \sigma_u)}{\sigma_u} = \frac{\phi(\mathbf{x}, \sigma_t)}{\sigma_t} - \int_t^u \frac{\dot{\sigma}_s}{\sigma_s^2} D(\phi(\mathbf{x}, \sigma_s), \sigma_s) \, ds. \quad (68)$$

Letting $u \to \infty$, we claim that the right-hand side converges. Indeed, the empirical data distribution $p_\star$ has compact support, meaning all the data points are confined in a bounded region of $\mathbb{R}^d$. Since the values of $D$ are weighted averages of the data points, it follows that $D$ is also bounded. Then the integrand $\frac{\dot{\sigma}_s}{\sigma_s^2} D(\phi(\mathbf{x}, \sigma_s), \sigma_s)$ is absolutely integrable on $[t, \infty)$, hence the convergence follows. Moreover, the limit does not depend on $t$. Denote this limit by $\rho(\mathbf{x})$:

$$\boldsymbol{\rho}(\mathbf{x}) = \frac{\phi(\mathbf{x}, \sigma_t)}{\sigma_t} - \int_t^\infty \frac{\dot{\sigma}_s}{\sigma_s^2} D(\phi(\mathbf{x}, \sigma_s), \sigma_s) \, ds. \quad (69)$$

As shown in the previous part, we know that $D(\mathbf{x}, t) = c + o(1)$ as $t \to \infty$ with $c = \mathbb{E}[x_\star]$. Then, multiplying both sides of Equation (69) by $\sigma_t$ and rearranging, we have, for large $t$,

$$\phi(\mathbf{x}, \sigma_t) = \sigma_t \boldsymbol{\rho}(\mathbf{x}) + \sigma_t \int_t^\infty \frac{\dot{\sigma}_s}{\sigma_s^2} D(\phi(\mathbf{x}, \sigma_s), \sigma_s) \, ds \quad (70)$$

$$= \sigma_t \boldsymbol{\rho}(\mathbf{x}) + (c + o(1)) \sigma_t \int_t^\infty \frac{\dot{\sigma}_s}{\sigma_s^2} \, ds \quad (71)$$

$$= \sigma_t \boldsymbol{\rho}(\mathbf{x}) + c + o(1). \quad (72)$$

Since $\phi$ is a bijection, the above relation tells that $\boldsymbol{\rho}(\mathbf{x})$ is also a bijection. Next, we replace $\mathbf{x} \leftarrow \hat{\mathbf{x}}_t$ in the equation defining $\boldsymbol{\rho}(\mathbf{x})$, Equation (69), to obtain:

$$\boldsymbol{\rho}(\hat{\mathbf{x}}_t) = \mathbf{z} + \frac{\mathbf{x}_\star}{\sigma_t} - \int_t^\infty \frac{\dot{\sigma}_s}{\sigma_s^2} \boldsymbol{D}(\phi(\hat{\mathbf{x}}_t, \sigma_s), \sigma_s) \, \mathrm{d}s. \tag{73}$$

Since $\boldsymbol{\rho}$ is invertible, applying $\boldsymbol{\rho}^{-1}$ to both sides yields

$$\hat{\mathbf{x}}_t = \boldsymbol{\rho}^{-1} \left( \mathbf{z} + \frac{\mathbf{x}_\star}{\sigma_t} - \int_t^\infty \frac{\dot{\sigma}_s}{\sigma_s^2} \boldsymbol{D}(\phi(\hat{\mathbf{x}}_t, \sigma_s), \sigma_s) \, \mathrm{d}s \right) \tag{74}$$

$$= \boldsymbol{\rho}^{-1} \left( \frac{\tilde{\mathbf{x}}_t}{\sigma_t} + \frac{\mathbf{x}_\star - \hat{\mathbf{x}}_t}{\sigma_t} - \int_t^\infty \frac{\dot{\sigma}_s}{\sigma_s^2} \boldsymbol{D}(\phi(\hat{\mathbf{x}}_t, \sigma_s), \sigma_s) \, \mathrm{d}s \right) \tag{75}$$

Since all of $\mathbf{x}_\star$, $\hat{\mathbf{x}}_t$, and $\boldsymbol{D}$ are bounded by the largest norm of the data point, they are all finite. Hence, the last line shows that $\hat{\mathbf{x}}_t = \boldsymbol{\rho}^{-1}\left(\frac{\tilde{\mathbf{x}}_t}{\sigma_t} + \mathcal{O}(\frac{1}{\sigma_t})\right)$, demonstrating that $\hat{\mathbf{x}}_t$ is almost a deterministic function of $\tilde{\mathbf{x}}_t$. Therefore, $\mathbb{E}[\hat{\mathbf{x}}_t | \tilde{\mathbf{x}}_t] \approx \hat{\mathbf{x}}_t$ as required. Consequently, $\tilde{\mathcal{R}}_{t,\mathrm{GC}}$ satisfies

$$\tilde{\mathcal{R}}_{t,\mathrm{GC}} \ll \left( \frac{\dot{\sigma}_t}{\sigma_t} \right)^2. \tag{76}$$

This proves that $\tilde{\mathcal{R}}_{t,\mathrm{GC}} \ll \tilde{\mathcal{R}}_{t,\mathrm{IC}}$ as required. $\qquad\square$

## A.3 Transport Cost

As a base for the two corollaries presented in the paper, we will first derive a useful representation of the derivative of the transport cost.

The main purpose of the lemma is to provide a more tractable representation of $c'(t)$, the time derivative of the expected transport cost. We expect $c(t)$ to decrease with $t$ because the predicted data point $f(\mathbf{x}_t, \sigma_t)$ becomes more dependent on the noise $\mathbf{z}$ as $t$ increases. However, directly analyzing $f(\mathbf{x}_t, \sigma_t) - \mathbf{z}$ is challenging because the dependence of $f(\mathbf{x}_t, \sigma_t)$ on $\mathbf{z}$ is not explicit. Therefore, the lemma aims to:

- identify a quantity that better captures the dependence between $\mathbf{z}$ and $\mathbf{x}_t$;
- relate $c(t)$ to this quantity.

The proof proceeds by deriving a key property of the ground-truth consistency map $f$: it satisfies the transport equation,

$$\frac{\partial f}{\partial \sigma}(\mathbf{x}, \sigma_t) \, \dot{\sigma}_t + \frac{\partial f}{\partial \mathbf{x}}(\mathbf{x}, \sigma_t) \cdot \mathbf{v}_t(\mathbf{x}) = 0. \tag{77}$$

This equation is equivalent to saying that the conditional expectation of the time derivative of $f(\mathbf{x}_t, \sigma_t)$ given $\mathbf{x}_t$ is zero:

$$\mathbb{E}\left[ \frac{\partial}{\partial t} f(\mathbf{x}_t, \sigma_t) \,\middle|\, \mathbf{x}_t \right] = 0. \tag{78}$$

By leveraging this property, we can simplify $c'(t)$ into an expression involving $\mathbf{w}_t = \mathbf{z} - \mathbb{E}[\mathbf{z} \mid \mathbf{x}_t]$, the residual between the true noise $\mathbf{z}$ and its prediction given $\mathbf{x}_t$. This residual captures the uncertainty in predicting $\mathbf{z}$ based on $\mathbf{x}_t$, allowing us to relate $c'(t)$ directly to the prediction accuracy of $f$.

**Lemma 1** (**Transport cost of GC coupling**). Assume that $\boldsymbol{f}$ is a continuously differentiable function representing the ground-truth consistency model, *i.e.* the flow of the PF-ODE induced by the diffusion process $\mathbf{x}_t$. Define $\mathbf{w}_t = \mathbf{z} - \mathbb{E}[\mathbf{z}|\mathbf{x}_t] = \frac{1}{\dot{\sigma}_t}(\dot{\mathbf{x}}_t - \mathbb{E}[\dot{\mathbf{x}}_t \mid \mathbf{x}_t])$. Then:

$$c'(t) = -2\dot{\sigma}_t \mathbb{E}\left[ \left\langle \frac{\partial \boldsymbol{f}}{\partial \mathbf{x}}(\mathbf{x}_t, \sigma_t) \cdot \mathbf{w}_t, \mathbf{w}_t \right\rangle \right]. \tag{79}$$

*Proof.* Note that the inverse flow $\boldsymbol{f}^{-1}(\mathbf{y}, \sigma_t)$ transports the initial point $\mathbf{y}$ at time $t = 0$ along the vector field $\mathbf{v}_t$ up to time $t$. Consequently, $\boldsymbol{f}^{-1}$ is a flow with the corresponding vector field $\mathbf{v}_t$:

$$\frac{\partial}{\partial t}[\boldsymbol{f}^{-1}(\mathbf{y}, \sigma_t)] = \mathbf{v}_t(\boldsymbol{f}^{-1}(\mathbf{y}, \sigma_t)). \tag{80}$$

By differentiating both sides of the identity $\mathbf{y} = \boldsymbol{f}(\boldsymbol{f}^{-1}(\mathbf{y}, \sigma_t), \sigma_t)$ with respect to $t$ and applying the above observation, we get:

$$0 = \frac{\partial}{\partial t}\left[\boldsymbol{f}(\boldsymbol{f}^{-1}(\mathbf{y}, \sigma_t), \sigma_t)\right] \tag{81}$$

$$= \frac{\partial \boldsymbol{f}}{\partial \sigma}(\boldsymbol{f}^{-1}(\mathbf{y}, \sigma_t), \sigma_t)\dot{\sigma}_t + \frac{\partial \boldsymbol{f}}{\partial \mathbf{x}}(\boldsymbol{f}^{-1}(\mathbf{y}, \sigma_t), \sigma_t) \cdot \frac{\partial}{\partial t}[\boldsymbol{f}^{-1}(\mathbf{y}, \sigma_t)] \tag{82}$$

$$= \frac{\partial \boldsymbol{f}}{\partial \sigma}(\mathbf{x}, \sigma_t)\dot{\sigma}_t + \frac{\partial \boldsymbol{f}}{\partial \mathbf{x}}(\mathbf{x}, \sigma_t) \cdot \mathbf{v}_t(\mathbf{x}), \tag{83}$$

where the substitution $\mathbf{x} = \boldsymbol{f}^{-1}(\mathbf{y}, \sigma_t)$ is used in the last step. Consequently,

$$c'(t) = 2\mathbb{E}\left[\left\langle \frac{\partial}{\partial t}[\boldsymbol{f}(\mathbf{x}_t, \sigma_t)], \boldsymbol{f}(\mathbf{x}_t, \sigma_t) - \mathbf{z} \right\rangle\right] \tag{84}$$

$$= 2\mathbb{E}\left[\left\langle \frac{\partial \boldsymbol{f}}{\partial \sigma}(\mathbf{x}_t, \sigma_t)\dot{\sigma}_t + \frac{\partial \boldsymbol{f}}{\partial \mathbf{x}}(\mathbf{x}_t, \sigma_t) \cdot \dot{\mathbf{x}}_t, \boldsymbol{f}(\mathbf{x}_t, \sigma_t) - \mathbf{z} \right\rangle\right] \tag{85}$$

$$= 2\mathbb{E}\left[\left\langle \frac{\partial \boldsymbol{f}}{\partial \mathbf{x}}(\mathbf{x}_t, \sigma_t) \cdot (\dot{\mathbf{x}}_t - \mathbf{v}_t(\mathbf{x})), \boldsymbol{f}(\mathbf{x}_t, \sigma_t) - \mathbf{z} \right\rangle\right] \tag{86}$$

$$= 2\dot{\sigma}_t\mathbb{E}\left[\left\langle \frac{\partial \boldsymbol{f}}{\partial \mathbf{x}}(\mathbf{x}_t, \sigma_t) \cdot (\mathbf{z} - \mathbb{E}[\mathbf{z}|\mathbf{x}_t]), \boldsymbol{f}(\mathbf{x}_t, \sigma_t) - \mathbf{z} \right\rangle\right], \tag{87}$$

where we used the relations $\mathbf{x}_t = \mathbf{x}_\star + \sigma_t\mathbf{z}$ and $\mathbf{v}_t(\mathbf{x}) = \mathbb{E}[\dot{\mathbf{x}}_t|\mathbf{x}_t]$. Now, let $\mathbf{w}_t = \mathbf{z} - \mathbb{E}[\mathbf{z} \mid \mathbf{x}_t]$. Then $\mathbb{E}[\mathbf{w}_t \mid \mathbf{x}_t] = 0$, hence by an application of the law of iterated expectations, $\mathbb{E}[\langle\mathbf{w}_t, g(\mathbf{x}_t)\rangle] = 0$ for essentially any function $g : \mathbb{R}^d \to \mathbb{R}^d$. Using this, we can further simplify the last line as:

$$c'(t) = -2\dot{\sigma}_t\mathbb{E}\left[\left\langle \frac{\partial \boldsymbol{f}}{\partial \mathbf{x}}(\mathbf{x}_t, \sigma_t) \cdot \mathbf{w}_t, \mathbf{z} \right\rangle\right] = -2\dot{\sigma}_t\mathbb{E}\left[\left\langle \frac{\partial \boldsymbol{f}}{\partial \mathbf{x}}(\mathbf{x}_t, \sigma_t) \cdot \mathbf{w}_t, \mathbf{w}_t \right\rangle\right], \tag{88}$$

proving the desired equality. $\qquad \square$

An immediate consequence of this lemma is that $c(t)$ decreases for small $t$:

**Corollary 1 (Decreasing transport cost of GC coupling in $t \to 0^+$).** There exists a $t_* > 0$ such that for all $t \in [0, t_*]$, the derivative of $c(t)$ takes the form $c'(t) = -2\dot{\sigma}_t a_t$ with $a_t > 0$. Hence for $\dot{\sigma}_t$ positive, the cost is decreasing. In particular, in the EDM setting where $\sigma_t = t$, $c(t)$ is decreasing for small $t$.

*Proof.* The proof of this corollary proceeds by noting that for $t = 0$, the consistency model $\boldsymbol{f}_\theta(\mathbf{x}, t)$ is an identity function, its Jacobian is an identity matrix leading to $a_t = \mathbb{E}[\|\mathbf{w}_t\|^2] > 0$ and by assumption, all the elements of the Jacobian are continuous. By continuity of $a_t$, $t_*$ exists and invoking intermediate value theorem on $a_t$ concludes the proof. $\qquad \square$

The next result is the statement about the asymptotic behavior of the transport cost $c(t)$ in the large-$t$ regime.

**Corollary 2 (Decreasing transport cost of GC coupling in $t \approx t_{\max}$).** Assume that the consistency model $\boldsymbol{f}_\theta(x, \sigma)$ is a scaling function $\boldsymbol{f}_\theta(\mathbf{x}, \sigma_t) = \frac{\sigma_0}{\sigma_t}\mathbf{x}$. Then, we have $c'(t) = -\frac{2\dot{\sigma}_t\sigma_0}{\sigma_t}\mathbb{E}[\|\mathbf{w}_t\|^2]$. In particular, $c(t)$ is decreasing whenever $\sigma_t$ is increasing.

*Proof.* Under the assumption, we have $\frac{\partial \boldsymbol{f}}{\partial \mathbf{x}} = \frac{\sigma_0}{\sigma_t}\mathbf{I}$. Thus, by Lemma 1,

$$c'(t) = -2\dot{\sigma}_t\mathbb{E}\left[\left\langle \frac{\sigma_0}{\sigma_t}\mathbf{I}\mathbf{w}_t, \mathbf{w}_t \right\rangle\right] = -\frac{2\dot{\sigma}_t\sigma_0}{\sigma_t}\mathbb{E}[\|\mathbf{w}_t\|^2]. \tag{89}$$

This proves that $c'(t) < 0$ whenever $\dot{\sigma}_t > 0$. $\qquad \square$

## B ADDITIONAL RESULTS

### B.1 ABLATION STUDIES

**Iso wall-clock training time.** As illustrated above, consistency models trained with GC converge faster than IC. However, each training step is more time-consuming, as it necessitates a forward evaluation of the consistency model without gradient computation. Regarding wall-clock training time, the computational overhead of iCT-GC is approximately 20% of the iCT-IC. In top part of Table 2, we report under "iCT-GC ($\mu = 0.5$) iso-time" the results of iCT-GC ($\mu = 0.5$) trained for as many hours as iCT-IC. Even when considering wall-clock training time, iCT-GC ($\mu = 0.5$) is still superior to iCT-IC.

Table 2: Analysis of performance with regards to some hyper-parameters of iCT-GC ($\mu = 0.5$) on CIFAR-10.

| Model | FID |
|---|---|
| iCT-IC | $7.42 \pm 0.04$ |
| iCT-GC ($\mu = 0.5$) iso-time | $\mathbf{6.38} \pm 0.03$ |
| iCT-GC ($\mu = 0.5$) | $\mathbf{5.95} \pm 0.05$ |
| iCT-GC($\mu = 0.5$) + dropout | $7.77 \pm 0.04$ |
| iCT-GC ($\mu = 0.5$) - EMA | $6.73 \pm 0.05$ |

**Hyper-parameters.** We evaluate the influence of two important hyper-parameters. First, the dropout in the learned model. Second, whether to use or not the EMA to compute GC endpoints $\hat{x}$. The results are presented in bottom part of Table 2. Interestingly, the results on dropout are opposite to those found by (Song and Dhariwal, 2024), since using dropout lowers the performance of iCT-GC ($\mu = 0.5$).

**Analysis of $\mu$ on ImageNet.** We present further results of the mixing procedure with varying $\mu$ ($\{0.3, 0.5, 0.7, 1.\}$) on ImageNet-32 in Figure 8. For $\mu = \{0.3, 0.5\}$, iCT-GC outperforms the base model iCT-IC.

### B.2 VISUAL RESULTS

We include in Figure 7 examples of generated images for considered baselines.

## C EXPERIMENTAL DETAILS

The code is based on the PyTorch library (Paszke et al., 2019).

**Scheduling functions and hyperparameters from Song and Dhariwal (2024).** The training of consistency models heavily rely on several scheduling functions. First, there is a noise schedule $\{\sigma_i\}_{i=0}^{N}$ which is chosen as in Karras et al. (2022). Precisely, $\sigma_i = \left(\sigma_0^{\frac{1}{\rho}} + \frac{i}{N}(\sigma_N^{\frac{1}{\rho}} - \sigma_0^{\frac{1}{\rho}})\right)^{\rho}$ with $\rho = 7$. Second, there is a weighting function that affects the training loss. It is chosen as $\lambda(\sigma_i) = \frac{1}{\sigma_{i+1} - \sigma_i}$. Combined with the choice of noise schedule, it emphasizes to be consistent on timesteps with low noise. Then, Song et al. (2023) proposed to progressively increase the number of timesteps $N$ during training. Song and Dhariwal (2024) argue that a good choice of dicretization schedule is an exponential one, which gives $N(k) = \min(s_0 2^{\lfloor \frac{k}{K'} \rfloor}, s_1) + 1$ where $K' = \lfloor \frac{K}{\log_2[s_1/s_0]+1} \rfloor$, $K$ is the total number of training steps, $k$ is the current training step, $s_0$ (respectively $s_1$) the initial (respectively final) number of timesteps. Finally, Song and Dhariwal (2024) propose a discrete probability distribution on the timesteps which mimics the continuous probability distribution recommended in the continuous training of score-based models by Karras et al. (2022). It is defined as $p(\sigma_i) \propto \text{erf}(\frac{\log(\sigma_{i+1}) - P_{\text{mean}}}{\sqrt{2}P_{\text{std}}}) - \text{erf}(\frac{\log(\sigma_i) - P_{\text{mean}}}{\sqrt{2}P_{\text{std}}})$. In practice, Song and Dhariwal (2024) recommend using: $s_0 = 10$, $s_1 = 1280$, $\rho = 7$, $P_{\text{mean}} = -1.1$, $P_{\text{std}} = 2.0$.

We use the lion optimizer (Chen et al., 2023) implemented from https://github.com/lucidrains/lion-pytorch.

**Selection of hyper-parameter $\mu$.** We have selected $\mu$ based on the results from Figure 6, which presents a grid search for $\mu$ on CIFAR-10. Given the bell-shaped relationship observed between

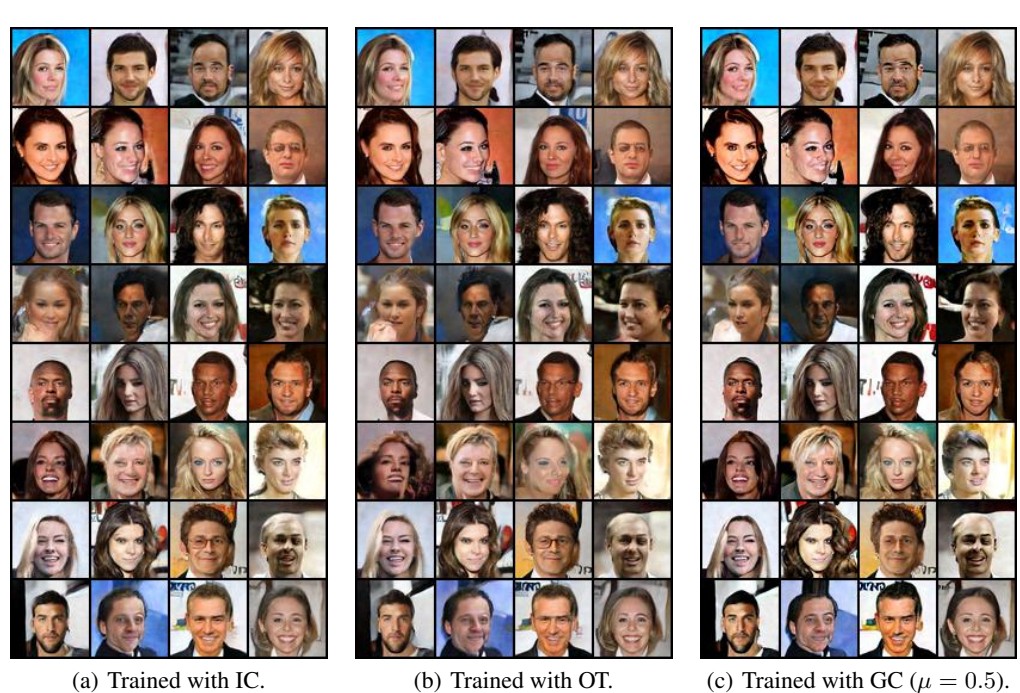

(a) Trained with IC.    (b) Trained with OT.    (c) Trained with GC ($\mu = 0.5$).

Figure 7: Uncurated samples from consistency models trained on CelebA $64 \times 64$ for fixed noise vectors. Note that models trained with generator-induced trajectories tend to generate sharper images.

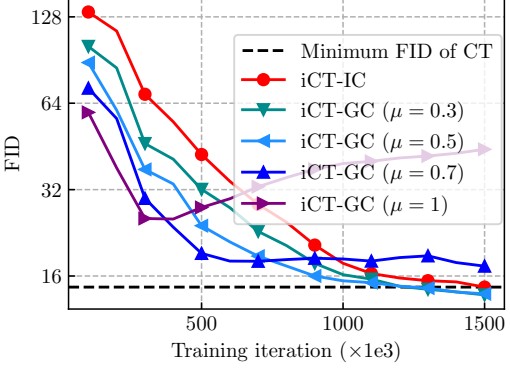

Figure 8: Results of varying $\mu$ for iCT-GC on ImageNet-32.

Table 3: Hyperparameters for CIFAR-10. Arrays indicate quantities per resolution of the UNet model. {} indicate an hyper-parameter search performed for each type of model (iCT, iCT-OT, iCT-GC ($\mu = 0.5$)).

| Hyperparameter | Value |
|---|---|
| batch size | 512 |
| image resolution | 32 |
| training steps | 100 000 |
| learning rate | $\{0.0001, 0.00003\}$ |
| optimizer | lion |
| $s_0$ | 10 |
| $s_1$ | 1280 |
| $\rho$ | 7 |
| $\sigma_0$ | 0.002 |
| $\sigma_1$ | 80 |
| network architecture | SongUNet |
| | (from Karras et al. (2022) implementation) |
| model channels | 128 |
| dropout | $\{0., 0.3\}$ |
| num blocks | 3 |
| embedding type | positional |
| channel multiplicative factor | $[1, 2, 2]$ |
| attn resolutions | $\emptyset$ |

$\mu$ and FID, we opted to retain the best performing value identified on CIFAR-10, $\mu = 0.5$, for all subsequent experiments (Table 1), including those on other datasets, without further tuning. Importantly, even without an exhaustive hyperparameter search, our method consistently outperforms baseline approaches. This choice is validated by the ablation study presented in Appendix B.1 showing similar trend for another dataset, showing that the hyper-parameter $\mu$ is easy to tune.

**Details on neural networks architectures.** We use the NCSN++ architecture (Song et al., 2021) and follow the implementation from https://github.com/NVlabs/edm.

**Evaluation metrics.** We report the FID, KID and IS. For the three different metrics, we rely on the implementation from TorchMetrics (Skafte Detlefsen et al., 2022). For the three different metrics, we use the standard practice (e.g. Song and Dhariwal (2024)) of FID which is to compare sets of 50 000 generated versus training images. Confidence intervals reported in Table 1 are averaged on five runs by sampling new sets of training images, and new sets of generated images from the same model.

**Datasets.** CIFAR-10 is a dataset introduced in Krizhevsky (2009). ImageNet (Deng et al., 2009), CelebA (Liu et al., 2015), and LSUN Church (Yu et al., 2015) are used respectively at $32 \times 32$, $64 \times 64$ and $64 \times 64$ resolutions. We preprocess these images by resizing smaller side to the desired value, center cropping, and linearly scaling pixel values to $[-1, 1]$.

**Details on computational ressources** As mentioned in the paper, the image dataset experiments have been conducted on NVIDIA A100 40GB GPUs.

# D BROADER IMPACTS

If used in large-scale generative models, notably in text-to-image models, this work may increase potential negative impacts of deep generative models such as *deepfakes* (Fallis, 2021).

Table 4: Hyperparameters for CelebA and LSUN Church. Arrays indicate quantities per resolution of the UNet model. {} indicate an hyper-parameter search performed for each type of model (iCT, iCT-OT, iCT-GC ($\mu = 0.5$)).

| Hyperparameter | Value |
| --- | --- |
| batch size | 128 |
| image resolution | 64 |
| training steps | 150 000 |
| learning rate | 0.00008 |
| optimizer | lion |
| $s_0$ | 10 |
| $s_1$ | 1280 |
| $\rho$ | 7 |
| $\sigma_0$ | 0.002 |
| $\sigma_1$ | 80 |
| network architecture | SongUNet |
| | (from Karras et al. (2022) implementation) |
| model channels | 128 |
| dropout | $\{0., [0., 0., 0.2, 0.2]\}$ |
| num blocks | $[3, 3, 4, 5]$ |
| embedding type | positional |
| channel multiplicative factor | $[1, 2, 2, 2]$ |
| attn resolutions | $\emptyset$ |

Table 5: Hyperparameters for ImageNet-1k. Arrays indicate quantities per resolution of the UNet model. {} indicate an hyper-parameter search performed for each type of model (iCT, iCT-OT, iCT-GC ($\mu = 0.5$)).

| Hyperparameter | Value |
| --- | --- |
| batch size | 512 |
| image resolution | 32 |
| training steps | 150 000 |
| learning rate | 0.00008 |
| optimizer | lion |
| $s_0$ | 10 |
| $s_1$ | 1280 |
| $\rho$ | 7 |
| $\sigma_0$ | 0.002 |
| $\sigma_1$ | 80 |
| network architecture | SongUNet |
| | (from Karras et al. (2022) implementation) |
| model channels | 128 |
| dropout | $\{0., [0., 0., 0.2, 0.2]\}$ |
| num blocks | $[3, 5, 7]$ |
| embedding type | positional |
| channel mult | $[1, 1, 2]$ |
| attn resolutions | $[16]$ |

