# OpenReview forum: "Improving Consistency Models with Generator-Induced Flows"
_ICLR.cc/2025/Conference — Submitted to ICLR 2025_

### Official Review · Reviewer_LCWL · 2024-11-03

**Soundness:** 3
**Presentation:** 3
**Contribution:** 3
**Rating:** 8
**Confidence:** 4

**Summary:**

The paper gave a negative answer to the  resorbent in the continuous-time limit of the discrepancy between consistency distillation
and training. Further, they had a new disovery that the an induced flow inside the generator, which could be used to train a consistent model.
They conducted great work on verifying their discovery in theory.

**Strengths:**

A very nice work on observation and anlysis  that showed some very interesting insight on CM. It remove an unclear fact about the CMs, which will be very useful for later discussions.

**Weaknesses:**

The experiments are not extensive. To make the observation and analysis more convining, the authors may try more datesets with different iamge sizes, and more imange generation metrics as those in paper in Ho 2020, Song 2021, Karras 2022, LIu 2023 et. al.

**Questions:**

It will be better to give some  high-level intuitive explanation of the key ideas behind the mathematics in the proofs; e.g., Lemma 1 and its restatement

---

> ### Author Response · Authors · 2024-11-18
>
> We would like to thank the reviewer for their positive assessment of our work. We appreciate your recognition of our theoretical contributions and insights regarding consistency models.
>
> ### Additional datasets
>
> > *(Weakness)* The experiments are not extensive. To make the observation and analysis more convining, the authors may try more datesets with different iamge sizes, and more imange generation metrics as those in paper in Ho 2020, Song 2021, Karras 2022, LIu 2023 et. al.
>
> Given our computational constraints, we hope the reviewer understands that we are unable to perform experiments at the same scale as the cited articles. Still, we highlight that our experiments were conducted on as many datasets as most of the cited papers, including the standard benchmarks of CIFAR-10 and ImageNet.
>
>
> ### Proof intuition
>
> > *(Question)* It will be better to give some high-level intuitive explanation of the key ideas behind the mathematics in the proofs; e.g., Lemma 1 and its restatement
>
> We thank the reviewer for the relevant suggestion. We agree that Lemma 1 may seem technical without explanation and that providing more intuitive insight would be beneficial. **We added the following explanation in Appendix Section A.3 with the revision of our submission.**
>
> The main purpose of the lemma is to provide a more tractable representation of $c'(t)$, the time derivative of the expected transport cost. We expect $c(t)$ to decrease with $t$ because the predicted data point $f(\mathbf{x}_t, \sigma_t)$ becomes more dependent on the noise $\mathbf{z}$ as $t$ increases. However, directly analyzing $f(\mathbf{x}_t, \sigma_t) - \mathbf{z}$ is challenging because the dependence of $f(\mathbf{x}_t, \sigma_t)$ on $\mathbf{z}$ is not explicit. Therefore, the lemma aims to:
>
> 1. identify a quantity that better captures the dependence between $\mathbf{z}$ and $\mathbf{x}_t$;
> 2. relate $c(t)$ to this quantity.
>
> The proof proceeds by deriving a key property of the ground-truth consistency map $f$: it satisfies the transport equation,
>
> $$
> \frac{\partial f}{\partial \sigma}(\mathbf{x}, \sigma_t) \, \dot{\sigma}_t + \frac{\partial f}{\partial \mathbf{x}}(\mathbf{x}, \sigma_t) \cdot \mathbf{v}_t(\mathbf{x}) = 0.
> $$
>
> This equation is equivalent to saying that the conditional expectation of the time derivative of $f(\mathbf{x}_t, \sigma_t)$ given $\mathbf{x}_t$ is zero:
>
> $$
> \mathbb{E}\left[ \frac{\partial}{\partial t} f(\mathbf{x}_t, \sigma_t) \,\middle|\, \mathbf{x}_t \right] = 0.
> $$
>
> By leveraging this property, we can simplify $c'(t)$ into an expression involving $\mathbf{w}_t = \mathbf{z} - \mathbb{E}[\mathbf{z} \mid \mathbf{x}_t]$, the residual between the true noise $\mathbf{z}$ and its prediction given $\mathbf{x}_t$. This residual captures the uncertainty in predicting $\mathbf{z}$ based on $\mathbf{x}_t$, allowing us to relate $c'(t)$ directly to the prediction accuracy of $f$.

---

> > ### Comment · Reviewer_LCWL · 2024-11-26
> >
> > Thanks. I am satisfied with your explanation.

---

### Official Review · Reviewer_obag · 2024-11-06

**Soundness:** 1
**Presentation:** 2
**Contribution:** 2
**Rating:** 3
**Confidence:** 4

**Summary:**

This paper studies the estimation error in the single-sample Monte Carlo estimates in the consistency training. The authors show that in the continuous-time limit, there exists a discrepancy between consistency training and consistency distillation (consistency distillation uses a pre-trained model to sample adjacent points instead of a single-sample Monte Carlo estimate. To remedy this issue, the authors propose generator-induced flows, which introduce a correlation between noise and the data at $t=0$. Experiments on image datasets show that the coupling leads to improved performance.

**Strengths:**

- The paper studies a new way of sampling the "clean" data in consistency training. The clean data $\hat{x}_0$ are online predictions of the current neural network, given $x_t \sim p_t$ , where $x_t=x_0+t z, x_0 \sim p_0, z \sim N(0,I)$. In this way, it introduces a "generator-induced coupling" (GC) between $\hat{x}_0$ and $z$

- The paper studies the distribution shift between independent coupling in the original consistency training and the GC. The authors propose a mix-training strategy to overcome this issue.

- Experimental results show that the mix-training strategy gives best results across image datasets.

**Weaknesses:**

- The paper has severe flaws in the main theorem (theorem 1) and the algorithm. I kindly request the authors to address them carefully or correct me if I'm wrong:

1.  The time derivative of the CT model (Eq.9 in theorem 1) is incorrect. Indeed, when the CT converges, $\dot{x}_t = v_t(x_t)$, as the CT will learn ground-truth ODE instead of "single-sample estimate" (please refer to the proof in the original consistency model paper). The idea is very similar to how the denoising score-matching objective works: the posterior of the "single-sample estimate" gives a ground-truth score, and CT is theoretically sound in the continuous-time limit. Also, the authors keep referring to the time derivative $\dot{x}_t$ as a "single-sample estimate", which is quite confusing. Could the authors crisply state the exact mathematical formulation of $\dot{x}_t$ in the theorem?

2. As the theorem is incorrect, so is Algorithm 1. Think of a counter-example in Algorithm 1: the current online model $f_\theta$ only maps all the intermediate points $x_t$ to a certain class, say "dog" class. As a result, the model will only learn to generate the doggy images since the algorithm replaces the clean data with the generated one by $f_\theta$. This scenario is very likely, as the initial $f_\theta$ likely drops many modes in the data distribution when $t$ is relatively large --- once it drops the modes, these lost modes will never be recovered.

3. The observations in Fig.4a and Fig.4b can be easily explained by the point 2 above. The authors utilize a hacky way (mix-training) to fix this.


- Despite the flaws discussed above, the proposed algorithm seems to improve over baseline on several image datasets. I think there are two possible reasons:

1. The current FID is in a poor regime, and the baselines are not well-tuned. For example, the iCT model can achieve a FID < 3 on CIFAR-10, instead of >7 in the paper. These poor FIDs make the results look unliable. In addition, could the authors test the results on ImageNet-64? This dataset is more representative than the datasets used in the paper.

2. There are some other benefits of using the generator-induced clean data. For example, the online $f_\theta$ could potentially "purify" the samples, e.g., make the samples more smooth. These purified data distributions at $t=0$ are easier to learn in the CT setting. It's analogous to training a model with soft labels instead of one-hot labels. I wonder what the authors' view on this is.


Due to the current status, I'm leaning toward rejecting the paper unless the authors clarify some of the critical points above.

**Questions:**

Please see the weakness section above.

---

> ### Author Response · Authors · 2024-11-18
> **Answer (1/2)**
>
> We would like to thank the reviewer for their constructive review and for recognizing the novelty and empirical value of our improvement of consistency models.
>
> The reviewer's main concern is the soundness of our theoretical results and of our algorithm. We provide a detailed response which should clarify the raised issues. As a consequence, we hope this also alleviates the reviewer's concerns on our experimental results, which we address in a second time. We look forward to further discussing with the reviewer to clarify any remaining concern.
>
> ## Soundness
>
> ### Theorem & $\dot{x}_t \neq v_t(x_t)$
>
> > 1. The time derivative of the CT model (Eq.9 in theorem 1) is incorrect. Indeed, when the CT converges, $\dot{x}_t = v_t(x_t)$, as the CT will learn ground-truth ODE instead of "single-sample estimate". [...] Also, the authors keep referring to the time derivative $\dot{x}_t$ as a "single-sample estimate", which is quite confusing. Could the authors crisply state the exact mathematical formulation of $\dot{x}_t$ in the theorem?
>
> We appreciate this important question and opportunity to clarify our paper. The raised issue originates in a confusion between $\dot{x}_t$ -- the time derivative of the *random variable* $x_t$ --, and $v_t(x_t)$, the time derivative of *particles* $x_t$ in the PF-ODE of Equation (1). Song et al. (2023), who adopt the same notations, go from distillation to training by replacing the latter by the former. Their difference explains the discrepancy outlined in our theorem. Therefore, **our theorem is correct**.
>
> Following the reviewer's suggestion, we explicitly defined $\dot{x}_t$ in the submitted revision (Line 127). We apologize for the confusion in the notations. Let us clarify in this response as well the mathematical formulation and re-define the key quantities involved.
> 1. The random variable $x_t$ is defined on Line 104: $x_t = x_\star + \sigma_t z$, where $x_\star \sim p_\star$ (sample from data distribution), $z\sim p_z$ (sample from the Gaussian), and $\sigma_t$ is monotonically increasing for $t\in[0,T]$.
> 2. Therefore, the time derivative $\dot{x}\_t$ is $\dot{x}\_t = \frac{\mathrm{d}(x_\star + \sigma_t z)}{\mathrm{d}t} = \frac{\mathrm{d} \sigma_t}{\mathrm{d}t}z=\dot{\sigma}_t z$. In the EDM case where $\sigma_t=t$, this simplifies to $\dot{x}_t = z$.
> 3. The velocity field $v_t$ is defined as: $v_t(x) = \mathbb{E}[\dot{x}_t | x_t=x]$ on Line 127.
>
> This shows that **$\dot{x}_t$ is indeed a single-sample estimate of $v_t(x_t)$**, and should not be confused with the time derivative in the ODE (Eq. 1), which is $v_t$ itself.
>
> ### Algorithm failure case
>
> > 2. As the theorem is incorrect, so is Algorithm 1. Think of a counter-example in Algorithm 1: if the current online model $f_\theta$ only maps all the intermediate points $x_t$ to a certain mode, then all the other modes will never be recovered.
>
> We thank the reviewer for the interesting remark.
>
> First, let us notice that **this failure case is shared with standard consistency models**. Indeed, in a standard consistency model, the performance of the model at timestep $i$ depends on the performance of the model at timestep $i-1$. If a model drops a mode during training at timestep $i-1$, the mode will not be recovered at timestep $i$, since it learns by following the output of the current model at $i-1$. Second, note that the consistency models framework, which we share, tends to avoid this thanks to 1) a parameterization enforcing the preservation of data at timestep 0; 2) the consistency loss ensuring propagation of data from timestep $i-1$ to timestep $i$.
>
> In our specific setting of GC models, the use of the endpoint predictor could indeed be an additional source of mode collapse, given the poor performance of the endpoint predictor on IC trajectories discussed in Section 5.1. We show, however, that our **simple mixing strategy prevents this issue** in Section 5.3, as partial IC training can compensate any mode collapse from GC.
>
> ## Experiments
>
> ### Performance of baselines
>
> > The current FID is in a poor regime, and the baselines are not well-tuned.
>
> Note that the baseline is the standard iCT-IC. The difference between the results from the iCT paper and our baseline comes from computational constraints. While the original iCT paper used greater computational resources, our experiments maintain consistent settings across all methods (iCT-IC, iCT-GC, iCT-OT) for **fair comparison**, using identical model sizes, batch sizes, and training steps -- adapted to our computational constraints. Even with these constraints, our complete experimental suite required approximately 100 days of computation on A100 40GB GPUs. Matching the computational scale of the original iCT paper is unfortunately beyond our current resources.

---

> ### Author Response · Authors · 2024-11-18
> **Answer 2/2**
>
> ### Additional dataset
>
> > In addition, could the authors test the results on ImageNet-64?
>
> We conducted our experiments on ImageNet-32, which uses the same base dataset as ImageNet-64 but with images downsampled to 32×32 resolution rather than 64×64 because of computational constraints. The only difference between these datasets is the image resolution. Note that we include results on the 64×64 resolution with the CelebA and LSUN Church datasets.
>
> ### Generator purification effect
>
> > There are some other benefits of using the generator-induced clean data. For example, the online $f_\theta$ could potentially "purify" the samples, e.g., make the samples more smooth. These purified data distributions at $t=0$ are easier to learn in the CT setting. It's analogous to training a model with soft labels instead of one-hot labels. I wonder what the authors' view on this is.
>
> We thank the reviewer for the interesting and plausible hypothesis that aligns with some of our observations. When analyzing GC-only models, we observed that as the number of timesteps increases and performance degrades, the generated images tend to become overly smooth and simplistic -- essentially generating the "center" of each mode rather than experiencing mode collapse.
>
> However, beyond this "purification" effect, our analysis suggests that the key factors driving improved performance are the discrepancy $\mathcal{R}$ and the transport cost. This is supported by the success of iCT-OT, which achieves improved performance without any "data purification". The effect of OT is solely to modify the transport cost and, consequently, the discrepancy term $\mathcal{R}$. We validate this further in Figure 2 of the paper, where we compare the discrepancy across IC, batch-OT (in the new revision), and GC approaches. Thus, while data purification may play a role in GC's performance, our theoretical analysis provides a more comprehensive explanation that accounts for the success of both GC and OT methods.

---

> ### Author Response · Authors · 2024-11-25
>
> Dear Reviewer obag,
>
> We have carefully addressed your concerns in our initial response. We would be grateful if you could acknowledge our response and participate in the discussion to share your thoughts on the points we addressed in the rebuttal. We are happy to provide further clarifications or additional details to resolve any remaining concerns. Thanks again for your involvement.

---

### Official Review · Reviewer_hFie · 2024-11-12

**Soundness:** 3
**Presentation:** 3
**Contribution:** 2
**Rating:** 5
**Confidence:** 3

**Summary:**

This paper, "Improving Consistency Models with Generator-Induced Flow," studies the coupling mechanism used in training consistency models. Consistency models are a recently developed class of generative models that learn to perform the multi-step sampling of score-based diffusion in a single forward pass.Instead of relying on sampling trajectories from a pre-trained diffusion model, consistency training uses stochastic interpolation between the data distribution and noise distribution, known as independent coupling (IC). This approach shows a discrepancy from consistency distillation. The authors analyze the differences between consistency training under IC and consistency distillation. To address the limitations of IC, the paper proposes a new type of flow called generator-induced flow, which relies on generator-induced coupling (GC) for training consistency models. This proposed generator-induced coupling accelerates convergence and enhances model performance compared to the base IC method and optimal transport approaches.

**Theoretical Results**

- This paper characterizes the discrepancy between consistency distillation and consistency training in values and gradients using different couplings.
- Under mild conditions, the loss values differ between consistency distillation and consistency training for distance functions other than the standard L2 loss.
- Though the limiting gradient in the continuous time coincides with the L2 metric, the gradient of the limiting loss values still differs due to the error term when using the independent coupling (IC).
- Authors show that the proposed GC could reduce the discrepancy between consistency training and consistency distillation.

**Algorithmic Improvements**

- The authors discuss potential couplings that could reduce the gap with the velocity field.
- The paper introduces generator-induced coupling (GC) that capitalizes consistency models to approximate the velocity field.

**Empirical Performance**

- The proposed GC adds an affordable cost to training while reducing the transport cost on CIFAR10 compared with batched optimal transport and independent coupling.
- The mixed strategy using IC and GC achieves improved FID compared with IC and batched OT baselines.

**Strengths:**

- The theorectical analysis investigates the discrepancy between consistency distillation and consistency training. This analysis is novel for understanding consistency models.
-  The proposed generator-induced coupling (GC) is cost-effective compared to other methods like optimal transport (OT).
- This paper is well-written. The logical flow from the consistency model definition to the theoretical results on the gradient discrepancy, followed by the generator-induced coupling (GC) for reduced transport cost.
- The analysis and experiments on the transport cost induced by different couplings are helpful for understanding.
- Key insights are clear and discussed in detail. For example, theorem 1 characterizes the discrepancies between consistency distillation and consistency training under IC. This is a clear motivation for the proposed method. The authors also discuss the implication of this analysis, showing how coupling could lead to different transport costs for consistency models.

**Weaknesses:**

- The paper acknowledges that consistency models trained exclusively on GC can suffer from distribution shift issues. Specifically, the marginal distribution shift between IC and GC during intermediate timesteps affects performance. Using a pre-trained consistency model through independent coupling could avoid divergence points to an inherent limitation but largely increase the computational cost.

- While the mixed IC-GC strategy helps mitigate this, the need for careful balancing complicates its application.

- The baseline used in the current draft is a relatively weak benchmark modified from iCT-IC. The mixed strategy of GC and IC improves upon this baseline only by a small margin. It is unclear whether these improvements would diminish when compared to the standard iCT-IC or iCT-OT. Including ReFlow [1] as a baseline coupling for comparison would strengthen the analysis. Additionally, conducting comparisons with efficient benchmarks like ECMs [1] for larger model sizes and comparing them with the standard iCT-IC would provide a better evaluation.

[1] Liu, X., Gong, C., & Liu, Q. (2022). Flow straight and fast: Learning to generate and transfer data with rectified flow. arXiv preprint arXiv:2209.03003. \
[2] Geng, Z., Pokle, A., Luo, W., Lin, J., & Kolter, J. Z. (2024). Consistency Models Made Easy. arXiv preprint arXiv:2406.14548.

While I acknowledge that this paper presents interesting insights into analyzing consistency models, the proposed generator-induced coupling is less attractive in practice and may suffer from distribution shifts. Therefore, I am not fully convinced of its practical impact at this stage.

**Questions:**

Some of these questions have been already mentioned in the weaknesses.

It would be helpful if the authors could discuss the following questions.

- Why does the distribution shift occur specifically at intermediate timesteps between IC and GC trajectories?
- When generator-induced coupling (GC) improves convergence speed, the model reaches saturation early during training. Is there a deeper analysis of why this phenomenon occurs?

---

> ### Author Response · Authors · 2024-11-18
>
> We would like to thank the reviewer for their constructive assessment of our work. We appreciate their recognition of the provided theoretical and empirical insights on consistency models and our improvements. We address the raised weaknesses and questions below.
>
> ### Hyperparameter sensitivity to circumvent distribution shifts
>
> > Specifically, the marginal distribution shift between IC and GC during intermediate timesteps affects performance. [...]
> >
> > While the mixed IC-GC strategy helps mitigate this, the need for careful balancing complicates its application.
>
> Fortunately, we found that **balancing the mixed IC-GC is easy**. In this paper, we have already conducted a hyperparameter grid search on mixing ratio $\mu$ using the CIFAR-10 dataset. Given the bell-shaped relationship observed between $\mu$ and FID, we opted to retain the best performing value identified on CIFAR-10, $\mu=0.5$, for all subsequent experiments, including those on other datasets, without further tuning. Importantly, even without an exhaustive hyperparameter search, our method consistently outperforms baseline approaches.
>
> In the new paper version, we included **a new experiment exploring $\mu$ on ImageNet-32** to further assess its generalizability (cf Figure 8 in Appendix). It consistently outperforms the base model for $\mu={0.3,0.5}$. We also detailed the previous explanation for our choice of $\mu$ in Appendix Section C.
>
> ### Performance of baselines
>
> > The baseline used in the current draft is a relatively weak benchmark modified from iCT-IC. The mixed strategy of GC and IC improves upon this baseline only by a small margin. It is unclear whether these improvements would diminish when compared to the standard iCT-IC or iCT-OT.
>
> Note that **the baseline is the standard iCT-IC**. The difference between the results from the iCT paper and our baseline comes from computational constraints. While the original iCT paper used greater computational resources, our experiments maintain consistent settings across all methods (iCT-IC, iCT-GC, iCT-OT) for fair comparison, using identical model sizes, batch sizes, and training steps -- adapted to our computational constraints. Even with these constraints, our complete experimental suite required approximately 100 days of computation on A100 40GB GPUs. Matching the computational scale of the original iCT paper is unfortunately beyond our current resources.
>
> ### Questions
>
> > Why does the distribution shift occur specifically at intermediate timesteps between IC and GC trajectories?
>
> This occurs because of the properties of the generator-induced trajectories, discussed around Line 292 and in Equation (16). On small timesteps $t$, the parameterization of the consistency model implies that $p(\tilde{x}_t) \approx p(x_t)$, and particularly $p(\tilde{x}_0) = p(x_0)$. On large timesteps, the signal-to-noise ratio is negligible, thus the distribution becomes close to the noise distribution, hence $p(\tilde{x}_T) \approx p(x_T) \approx p(\sigma_T z)$. This explains why the distribution shift mostly happens on intermediate timesteps.
>
> > When generator-induced coupling (GC) improves convergence speed, the model reaches saturation early during training. Is there a deeper analysis of why this phenomenon occurs?
>
> Before anything else, let us remind that the saturation is not observed when using mixing, while retaining a higher convergence speed than the baseline IC. When using GC-only trajectories however, the model indeed reaches saturation early during training. In this setting, we believe that main limiting factor is the the distribution shift between IC and GC trajectories, as validated by experiments in Section 5.2. Our mixing strategy successfully solves this issue, as discussed in the paper and in this response.

---

> ### Author Response · Authors · 2024-11-25
>
> Dear Reviewer hFie,
>
> We have carefully addressed your concerns in our initial response. We would be grateful if you could acknowledge our response and participate in the discussion to share your thoughts on the points we addressed in the rebuttal. We are happy to provide further clarifications or additional details to resolve any remaining concerns. Thanks again for your involvement.

---

> ### Author Response · Authors · 2024-12-01
>
> Dear Reviewer **hFie**,
>
> We would like to thank you for your valuable suggestion. Following your advice, we conducted an **additional experiment** using the efficient ECM setting described in Geng et al. [1]. This configuration enables high performance with only 1 GPU-hour of fine-tuning from a pre-trained denoising model.
>
> Using this setup, we compared iCT-IC and iCT-GC ($\mu=0.7$) trained with a 1-hour fine-tuning: iCT-IC achieves a 1-step FID of 7.37, while **iCT-GC ($\mu=0.7$) achieves a lower FID** of 6.41. These findings highlight the advantages of generator-induced flows, particularly in scenarios where rapid convergence is important.
>
> [1] Geng et al., Consistency Models Made Easy, arXiv:2406.14548.

---

### Official Review · Reviewer_LmSt · 2024-11-12

**Soundness:** 3
**Presentation:** 4
**Contribution:** 4
**Rating:** 8
**Confidence:** 2

**Summary:**

This paper identifies a previously overlooked discrepancy between training and distillation in consistency models and introduces a novel approach called generator-induced flows to address it. This method improves model convergence speed and performance. Additionally, to tackle the distribution shift issue between independent coupling (IC) and generator-induced coupling (GC), the paper proposes a mixed training strategy, resulting in efficient and high-quality image generation.

**Strengths:**

- This paper is well-structured and easy to read. Each section clearly states its claims and conclusions, making it easy to understand.
- The Generator-Induced Coupling introduced by the authors is very straightforward and serves as a convincing method for resolving the discrepancy. Additionally, the paper's contribution is significant, as it points out aspects of the discrepancy that have been overlooked in previous studies.
- The proposed method surpasses the state-of-the-art iCT, demonstrating high practicality and effectiveness. Therefore, I believe this paper should be accepted.

**Weaknesses:**

The main difficulty of the proposed method in this study lies in the dilemma that, while aiming to reduce the discrepancy by predicting endpoints from the model, the model itself is still being trained and thus has weak performance as an endpoint predictor, leading to distribution shifts. To address this, the authors propose mixing IC and GC during training, but this solution is a compromise supported mainly by empirical evidence, and introduces a new challenge in determining the appropriate mixing ratio as a tunable parameter.

**Questions:**

- Could you please explain more about the "weak $g_\phi$ partially-trained as iCT-IC" mentioned in Section 5.2? Does this imply that the endpoint predictor is trained on both GC and IC, similar to the final proposed method?
- The performance of varying $\mu$ values on CIFAR-10 is shown in Figure 10, but what about the performance on other datasets with different mixing rates? I would also like to know how sensitive this parameter is for other models.

---

> ### Author Response · Authors · 2024-11-18
>
> We would like to thank the reviewer for their positive assessment of our work. We appreciate their recognition of our identification of the discrepancy term and the introduction of GC as a way to diminish it. We address the raised weaknesses and questions below.
>
> ### Distribution shifts
>
> > *(Weakness)* The main difficulty of the proposed method [is that] the model itself is still being trained and thus has weak performance as an endpoint predictor, leading to distribution shifts.
>
> We thank the reviewer for raising this important point. We want to clarify that the distribution shift is not primarily caused by the endpoint predictor's performance. Rather, it is an inherent characteristic of different probability paths ($p_t$) whether using GC or other trajectory-changing methods such as batch-OT. To illustrate this, consider Figure 1 in our paper where we use an optimal ground-truth consistency model. Even in this idealized case with perfect prediction, we observe distribution shifts (specifically, lower density around $x=0$ for GC compared to IC). Similarly, when using Optimal Transport to couple noise and data, the associated probability path differs from IC. These shifts arise from the different couplings generating different probability paths.
>
> ### "Weak partially-trained [predictor] as iCT-IC" in Section 5.2
>
> > *(Question)* Could you please explain more about the "weak partially-trained as iCT-IC" mentioned in Section 5.2? Does this imply that the endpoint predictor is trained on both GC and IC, similar to the final proposed method?
>
> Let us clarify the terminology in Section 5.2. We discuss two distinct models:
> - (i) A standard consistency model (iCT-IC) trained for 100k steps, which we call the IC predictor.
> - (ii) A standard consistency model (iCT-IC) trained for only 20k steps, which we refer to as the "weak partially-trained as iCT-IC" or weak-IC predictor.
>
> **These predictors are trained purely on IC**, not on both GC and IC. Using these models, we then conduct two experiments:
>
> - (i) Training a GC-only model using the fully-trained IC predictor (100k steps).
> - (ii) Training a GC-only model using the weak-IC predictor (20k steps).
>
> This approach allows us to compare how the endpoint predictor's performance affects the GC model's performance; cf. Finding 3 in Section 5.2:
> > the performance of the GC model depends on the quality of the endpoint predictor evaluated on IC trajectories.
>
> We specified this additional information in Section 5.2 of the new revision.
>
> ### Sensitivity of the mixing ratio hyperparameter
>
> > *(Weakness)* To address this, the authors propose mixing IC and GC during training, but this solution [...] introduces a new challenge in determining the appropriate mixing ratio as a tunable parameter.
>
> > *(Question)* The performance of varying values on CIFAR-10 is shown in Figure 10, but what about the performance on other datasets with different mixing rates? I would also like to know how sensitive this parameter is for other models.
>
> Fortunately, we found that **the mixing ratio $\mu$ is easy to tune** as a new hyper-parameter. In this version of the paper, we have already conducted a hyperparameter grid search on $\mu$ using the CIFAR-10 dataset. Given the bell-shaped relationship observed between $\mu$ and FID, we opted to retain the best performing value identified on CIFAR-10, $\mu=0.5$, for all subsequent experiments, including those on other datasets, without further tuning. Importantly, even without an exhaustive hyperparameter search, our method consistently outperforms baseline approaches.
>
> In the new paper version, we included **a new experiment exploring $\mu$ on ImageNet-32** to further assess its generalizability (cf Figure 8 in Appendix). It consistently outperforms the base model for $\mu={0.3,0.5}$. We also detailed the previous explanation for our choice of $\mu$ in Appendix Section C.

---

### Author Response · Authors · 2024-11-18
**Global answer**

We thank the reviewers for their positive and constructive feedback on our work. We provided initial responses, along with a revision of our submission, to initiate the discussion which we look forward to pursue.

We responded individually to each reviewer. Additionally, we would like to share to all the following information.

**Paper revision.**

Changes w.r.t. the initial submission are highlighted in blue.

1. As requested by **Reviewer obag**, we added a definition of $\dot{x}_t$ to remove the potential confusion between $\dot{x}_t$ and $v_t$. *(Line 127)*
2. We added $\tilde{\mathcal{R}}\_{\text{batch-OT}}$ in *Figure 2*. It further confirms the relevance of our analytical framework. Indeed, we see that $\tilde{\mathcal{R}}\_{\text{GC}} < \tilde{\mathcal{R}}\_{\text{batch-OT}} < \tilde{\mathcal{R}}\_{\text{IC}}$, which is the ordering that we observe on almost all datasets for final performance model (see Table 4).
3. As suggested by **Reviewer LCWL**, we added an high-level intuitive explanation of the proof of Lemma 1. *(Line 1000)*
4. As requested by **Reviewer LmST**, we added some details about the "weak partially-trained endpoint predictor". *(Line 476)*
5. Study for $\mu$ on ImageNet (**Reviewers LmST & hFie**): we added **a new experiment** in Appendix Figure 8. On CIFAR-10, the baseline iCT-IC is improved for $\mu={0.3,0.5,0.7}$. On ImageNet, it is improved for $\mu={0.3,0.5}$. This shows the stability of our performance w.r.t. to this new hyperparameter. *(Line 1103)*
6. Description of the rationale for the selection of $\mu=0.5$ (**Reviewers LmST & hFie**) in the Appendix. *(Line 1133)*

**Statement on experiments.** As noted by some reviewers, some of our baselines may appear weaker compared to those shown in the literature. This is primarily due to our limited access to high-end GPUs, which has constrained our ability to fully reproduce results from SOTA models as described in their original papers. We hope that this context will be taken in consideration, as we have fully complied to the scientific comparison methodology.

---

### Author Response · Authors · 2024-11-21

We sincerely thank the reviewers once again for their constructive feedback, which has been invaluable in improving and clarifying the paper. In particular, we addressed key concerns, such as the potential confusion between $\dot{x}_t$ and $v_t$ raised by **Reviewer obag**, as well as the remarks related to our empirical setting highlighted by **Reviewer hFie**, as detailed in our previous responses. We would be happy to provide further clarification on this or any other points if needed. Please let us know if there are additional questions or concerns.

---

### Author Response · Authors · 2024-12-01
**New experiment in ECM setting**

Dear Reviewers,

We would like to thank **Reviewer hFie** for their valuable suggestion. Following their advice, we conducted an **additional experiment** using the efficient ECM setting described in Geng et al. [1]. This configuration enables high performance with approximately 1 GPU-hour of fine-tuning (8k training steps) from a pre-trained denoising model.

Using this setup, we compared iCT-IC and iCT-GC ($\mu=0.7$) trained with a 1-hour fine-tuning: iCT-IC achieves a 1-step FID of 7.37, while **iCT-GC ($\mu=0.7$) achieves a lower FID** of 6.41. These findings highlight the advantages of generator-induced flows, particularly in scenarios where rapid convergence is important.

[1] Geng et al., Consistency Models Made Easy, arXiv:2406.14548.

---

### Public Comment · ~Thibaut_Issenhuth1 · 2025-02-06
**Response to Meta Review**

We would like to thank the reviewers and area chair for their detailed feedback and for acknowledging the promise of our work. Following the paper's rejection, we would like to respectfully address and dispute the three main concerns raised in the meta-review.

### 1. Discrepancy Between Theory and Practice

Reviewers LmSt and hFie regretted the empirical need to rely on a mixing strategy between IC and GC for stability, which is not an issue as we showed in our initial answer. In the area chair's assessment, mixing introduces a discrepancy betwen theory and practice, casting doubts on our claims. To address simultaneously both concerns, **we make clear in this message and in the new version of our paper ([https://arxiv.org/abs/2406.09570](https://arxiv.org/abs/2406.09570)) that mixing, on the contrary, stems naturally from our theory**.

The theoretical motivation for GC in our paper assumes access to an optimal generator on IC trajectories *(Theorem 2, Lemma 1 and its corollaries)*. However, when training with GC alone, the assumption of an ideal generator is broken. Consequently, our theory does not claim GC-only should succeed: **the divergence issue of GC alone does not raise concerns on the validity of our theoretical claims**.

Relying on an optimal, i.e. pre-trained in practice, IC generator is impractical and limits GC-based generator’s performance, as demonstrated in Section 5.2. To comply with the *theoretical requirement* of having an IC-generator with great performance, we propose a mixed coupling that combines training on trajectories from GC and IC. It is rooted in both practical constraints (one phase training) and theoretical considerations (optimal generator on IC). As a result, **the mixing strategy is motivated by our theoretical analysis**.

### 2. Scale of Experimental Validation

While we acknowledge that our experimental setup does not match the scale of some recent high-resource studies -- stated as an important concern by the area chair and Reviewer obag only --, we maintain that **it is sufficient to validate our claims** regarding convergence speed and improved performance.
- Despite constrained resources, we provide meaningful insights into the training dynamics of consistency models.
- We do so within reasonable FID values -- our IC baseline operating within the ranges of the original consistency model of [Song et al. (2023)](https://proceedings.mlr.press/v202/song23a.html) on CIFAR-10 -- and on four standard datasets -- more than several other consistency models: [Song & Dhariwal (2024)](https://openreview.net/forum?id=WNzy9bRDvG), [Lu & Song (2024)](https://arxiv.org/pdf/2410.11081), [Geng et al. (2024)](https://arxiv.org/pdf/2406.14548).
- Our experiments are methodologically sound within the investigated scope, reimplementing baselines for fair comparison, using identical model sizes, batch sizes, and training steps for all.

Regarding our constrained resources, the area chair's analogy with "social science experiment made on 10 persons" is **inappropriate**. In this analogy, claims are not verifiable because of lacking statistical significance. In our case, our claims are explicitly validated within the reasonable and robust experimental framework we present. We believe that this is valuable and can be of interest for the community to replicate on a larger scale and build upon.

### 3. Metrics

The area chair questioned our choice of comparison metrics. We acknowledge the known limitations of metrics like FID, KID, and IS, as highlighted in recent critiques. However, **these metrics remain standard for generative model evaluation** unlike their alternatives, and their widespread use enables comparisons with prior work. Some of the most impactful and recently published diffusion/consistency papers are based on FID, *e.g.* [Karras et al. (2022)](https://openreview.net/forum?id=k7FuTOWMOc7), [Song et al. (2023)](https://proceedings.mlr.press/v202/song23a.html), as well as [Lu & Song (2024)](https://arxiv.org/pdf/2410.11081) and [Geng et al. (2024)](https://arxiv.org/pdf/2406.14548) at the present ICLR edition. This choice of metrics should not be ground for rejection.

---

### Meta-Review · Area_Chair_uPW2 · 2024-12-12

**Metareview:**

This paper explores the coupling mechanisms used to train consistency models and proposes a new generator-induced coupling (GC) which is meant to address the limitations of independent couplings (IC). In practice, using GC alone leads to distribution shifts, so a mixed coupling involving GC and IC is recommended.

Reviewers noted discrepancies between the experimental results and practical recommendations, compared to the motivation and theoretical discussions given by the authors. In particular, the need to combine GC with IC (a method for which the authors point out various flaws and set out to fix) is unsatisfying. It raises concerns as to the validity of the author’s claims. Also, the small scale and non-comprehensive nature of the experiments leaves room for doubt as to the actual benefits of the GC method. The models trained by the authors achieve FID values that are well above those reported in previous works, which might imply that the models are not well-converged.

In my view, the experimental results and practical recommendations do not align with the motivation and theoretical discussions given by the authors, which raises concerns as to the validity of those claims. The small scale and non-comprehensive nature of the experiments leaves room for doubt as to the actual benefits of the GC method.

Other concerns from the reviewers were addressed, such as some concerns with the theoretical results. Generally the reviewers appreciated the ideas in the work. Overall, this work is promising, and I believe that based on the feedback given by reviewers, it can be improved to the quality required for acceptance in a comparable venue. However, those revisions are extensive enough to require further peer-review, so I am not recommending acceptance at ICLR.

**Additional Comments On Reviewer Discussion:**

The main points of concern raised by reviewers were: GC alone does not provide good performance, and the authors had to combine GC and IC, which is not motivated by their theoretical arguments; generally weak experiments using small scale data and poorly-tuned baselines.

The authors managed to clarify reviewer concerns about some theoretical results. However, the main proposal of the paper to combine GC and IC was unsatisfying and would constitute a major revision if changed at this point.

In response to concerns that experiments were not extensive enough, and models were not well-converged, the authors appealed multiple times to the fact that they had limited compute (spending in total “100 days of computation on A100 40GB GPUs”). I want to address this point in detail. While I am sympathetic to the author’s position, it stands that some scientific claims simply require more compute to be convincing. By analogy, if a group of scientists ran a social science experiment with 10 participants because they did not have funding to recruit more, their results would probably not achieve statistical significance; peer-review should not give those results a pass just because of budgetary constraints. I acknowledge the risk that scientific results in machine learning may trend towards being out of reach of more and more researchers globally based on arguments like this. However, scientific results in many fields come at great financial costs (e.g. as an extreme case, the ATLAS experiment at CERN cost billions of dollars and hence is not replicable by any other scientific collaboration globally), so this is a dilemma we are forced to confront.

Although not noted by reviewers specifically, the experimental conclusions were largely based on FID, and to some extent KID and IS, which are metrics that have been widely criticized in the research community. See for example Stein et al. “Exposing flaws of generative model evaluation metrics and their unfair treatment of diffusion models” NeurIPS 2023. The authors did not give justification for why their chosen metrics were appropriate for their specific tasks in light of these concerns.

---

### Decision · Program_Chairs · 2025-01-22

Reject